# QuIC: Quantum-Inspired Compound Adapters for Parameter Efficient Fine-Tuning

## Abstract

Scaling full finetuning of large foundation models strains GPU memory and training time. Parameter Efficient Fine-Tuning (PEFT) methods address this issue via adapter modules which update only a small subset of model parameters. In this work, we introduce Quantum-Inspired Compound Adapters (QuIC Adapters), a PEFT approach inspired from Hamming-weight preserving quantum circuits that can effectively finetune a model using less than $0.02\%$ memory footprint of the base model. QuIC adapters preserve pretrained representations by enforcing orthogonality in weight parameters, and have native deployment mechanisms on quantum computers. We test QuIC adapters by finetuning large language models like LLaMA and vision transformers on language, math, reasoning and vision benchmarks. In its first-order configuration, QuIC recovers the performance of existing orthogonal methods, while higher-order configurations enable substantial parameter compression (over $40\times$ smaller than LoRA) for a modest performance trade-off, unlocking applications in highly resource-constrained environments. Through ablation studies, we determine that combining multiple Hamming-weight orders with orthogonality and matrix compounding are essential for performant finetuning. Our findings suggest that QuIC adapters offers a promising direction for efficient finetuning of foundation models in resource-constrained environments.

## 1 Introduction

Pre-trained large foundation models such as BERT (Devlin et al., 2018), GPT-3 (et al., 2020), and Vision Transformers (Dosovitskiy, 2020) have achieved state-of-the-art results on various tasks. Fine-tuning these models on specific downstream tasks typically involves updating all model parameters but with a lower learning rate, which becomes computationally prohibitive as model sizes continue to grow into the billions of parameters. This challenge has spurred interest in Parameter-Efficient Fine-Tuning (PEFT) methods (Houlsby et al., 2019), which aim to adapt large foundation models to new tasks by updating only a small subset of parameters or introducing lightweight adaptation modules.

One of the most prominent PEFT techniques is Low-Rank Adaptation (LoRA) (Hu et al., 2021), which injects low-rank trainable matrices into transformer layers, significantly reducing the number of parameters that need to be updated. Other methods like Adapters (Houlsby et al., 2019), BitFit (Ben Zaken et al., 2022), and Prompt Tuning (Lester et al., 2021) have also demonstrated effectiveness in various settings. Recently, Orthogonal Fine-Tuning (OFT) (Qiu et al., 2023) and its 'Butterfly' specification (BOFT) (Liu et al., 2023) have been proposed to mitigate catastrophic forgetting of the pre-trained models during finetuning by applying orthogonal transformations. These methods have shown promising results in achieving a balance between performance and parameter efficiency.

While methods like LoRA and OFT significantly reduce parameters compared to full finetuning, a critical need remains for even greater efficiency in resource-constrained scenarios. Deploying personalized models on-device (e.g., smartphones or wearables), serving thousands of task-specific adapters simultaneously, or reducing bandwidth in federated learning all impose strict memory and storage budgets that even conventional PEFT methods can exceed (Kopiczko et al., 2024; Zhang et al., 2023; YEH et al., 2024). This motivates the development of methods capable of extreme compression, pushing the Pareto frontier of what is achievable with a minimal parameter budget.

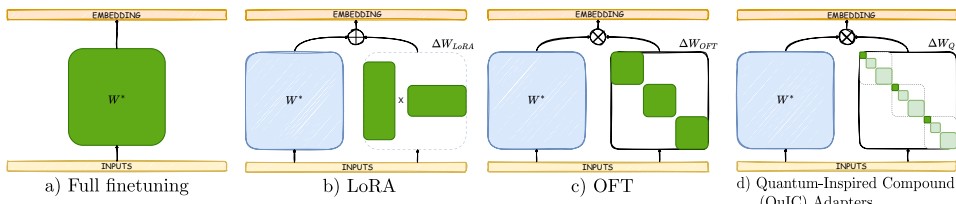

a) Full finetuning     b) LoRA     c) OFT     d) Quantum-Inspired Compound (QuIC) Adapters

Figure 1: **Comparison of different adapter methods.** Trainable parameters for each model shown in dark green. a) Full finetuning b) Low-rank adaptation (LoRA) c) Orthogonal finetuning (OFT) d) Quantum-Inspired Compound adapter (QuIC adapter). For QuIC adapters, the zeroth order compound (top left of each block) is the only trainable part. Higher order compounds are completely determined by this base matrix.

Inspired by the potential exponential compression abilities of quantum and quantum-inspired computing, there has been a growing interest in Quantum-Inspired PEFT methods such as QuanTA (Chen et al., 2024) and QPA (Liu et al., 2025). While QuanTA constructs adapters via contracted quantum-inspired tensor networks, Quantum Parameter Adaptation (QPA) uses quantum circuits to generate parameters for methods such as LoRA. These works highlight the potential for quantum machine learning within finetuning, however both methods contain a number of bottlenecks which potentially prohibit quantum computer integration with finetuning pipelines at larger scales.

In this work, we propose **Qu**antum-**I**nspired **C**ompound **Adapters** (QuIC Adapters), a novel PEFT method inspired by Hamming-weight preserving quantum circuits (Kerenidis & Prakash, 2022; Landman et al., 2022; Cherrat et al., 2023). With QuIC adapters, orthogonality is a native feature, and we focus on compound orders up to a certain constant $K$ to ensure parameter efficiency. We evaluate our method on several datasets over a variety of domains. For language, vision, reasoning and math problems, we use the the General Language Understanding Evaluation (GLUE) benchmark (Wang, 2018), a subset of tasks from the Visual Task Adaptation (VTAB) benchmark (Zhai et al., 2019), the Discrete Reasoning Over the text in the Paragraph (DROP) dataset (Dua et al., 2019), and the MATH10K (Hu et al., 2023) benchmark respectively. On the model side, we finetune the moderate size DeBERTaV3 (He et al., 2021) for language and DINOv2-large for vision. For a larger model and for math and reasoning tasks we focus on LLaMA-7B (Touvron et al., 2023b). Our experiments demonstrate that QuIC adapters achieve competitive performance while dramatically reducing the number of trainable parameters compared to existing PEFT methods like LoRA, OFT, BOFT and QuanTA, among others.

## 2 BACKGROUND

Large language and vision foundation models are largely based on the transformer architecture (Vaswani et al., 2017; Dosovitskiy, 2020; Devlin et al., 2018). In this section, we provide an overview of the core components of adapter based finetuning. These are primarily applied to attention and feedforward layers in a foundation model, and we give the explicit form in Appendix A. We also introduce Hamming-weight quantum machine learning, which serves as the inspiration for our approach.

### 2.1 PARAMETER-EFFICIENT FINE-TUNING METHODS

Generally speaking, PEFT methods finetune large pre-trained foundation models with layers $W^* \in \mathbb{R}^{d \times d}$ by training an *adapter* layer, denoted $\Delta W$. The PLM layers are then combined with the adapter to construct the finetuned model weight matrix, $W_{\text{adapt}}$. Then, PEFT methods are generally either *additive*, ($W_{\text{adapt}} := W^* + \Delta W$) or *multiplicative*, ($W_{\text{adapt}} := \Delta W \times W^*$).

Low-Rank Adaptation (LoRA) (Hu et al., 2021) is an additive adapter and has the form $\Delta W_{\text{LoRA}} := \alpha W_{\text{up}} W_{\text{down}}$ with $W_{\text{up}} \in \mathbb{R}^{d \times r}$, $W_{\text{down}} \in \mathbb{R}^{r \times d}$, and $\alpha$ is a scaling factor. The rank, $r$, of the trainable matrices, $W_{\text{up}}, W_{\text{down}}$ controls the number of trainable parameters and is typically $\ll d$.

On the other hand, (Butterfly) Orthogonal Fine-Tuning ((B)OFT) (Qiu et al., 2023; Liu et al., 2023) uses multiplicative adapters. (B)OFT adapters enforce an *orthogonality constraint*, i.e.

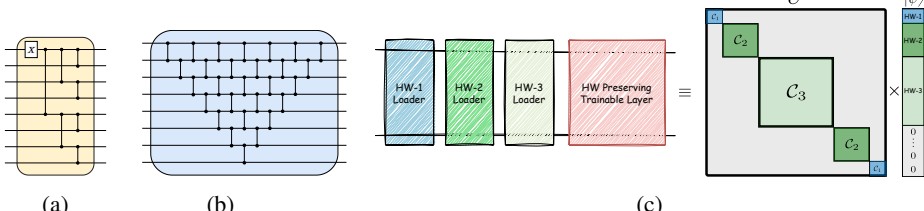

(a)                    (b)                                  (c)

Figure 2: **Hamming-weight preserving quantum computation.** Quantum circuits are read left to right and each vertical line corresponds to a Reconfigurable/Fermionic Beam Splitter (RBS/FBS) quantum gate with parameter $\theta$. a) A unary (parallel) data loader (Landman et al., 2022) to load a vector, $\mathbf{x}$, into Hamming-weight (HW) $k = 1$ states. Generalizations of such loaders to higher HW can be found (Farias et al., 2024) and discussed in the Appendix. b) A 'pyramid' trainable quantum circuit layer, which is HW preserving (Landman et al., 2022). c) The generalization into HW up to $K = 3$ states. The action of a HW preserving layer composed of FBS gates is represented by a unitary, $U$, composed of compound matrices, $\{\mathcal{C}_k := A^{(k)}\}$ acting on a data encoded state, $|\psi\rangle$. The elements of the vector representation of $|\psi\rangle$ are ordered according to Hamming-weight, and the compound matrices, $\mathcal{C}_k$, act separately on each set of HW grouped basis states. The matrices, $U$, themselves will serve as the inspiration for our QuIC Adapters.

$\Delta W_{\text{OFT}}^\top \Delta W_{\text{OFT}} = \mathbb{1}$ which ensures that the transformation preserves the spectral properties of $W^*$ and retains the pre-trained knowledge during finetuning. Different parameterizations of $\Delta W_{\text{OFT}}$ are possible - specifically, (Qiu et al., 2023) chooses to employ the Cayley transform (explicit equation given in Eqn. (4) in Appendix A). In OFT, further sparsity is enforced with a 'rank' parameter - controlling the block size across a block diagonal decomposition. Specifically, a block, $i$, is defined as an orthogonal matrix of size $\Delta W_{\text{OFT},i} \in \mathbb{R}^{d/r \times d/r}$. BOFT (Liu et al., 2023) extends OFT by introducing an efficient parameterization of the orthogonal matrix as a product of $m$ sparse orthogonal matrices derived from '*butterfly*' structures, $\Delta W_{\text{BOFT}} := \prod_{i=1}^m \widetilde{B}_{(i)}$, where each $\widetilde{B}_{(i)} \in \mathbb{R}^{d \times d}$ is a butterfly *factor* - a sparse orthogonal matrix, defined recursively, that efficiently captures global interactions within the data.

Finally, quantum-inspired finetuning methods such as QuanTA (Chen et al., 2024) build adapter matrices, $\Delta W$, via a contraction of *tensor networks* (TNs) - connected graphs of multi-dimensional tensorial objects motivated from attempts to study many body quantum systems using low-dimensional representations. These are inspired from general quantum circuits. On the other hand, Quantum Parameter Adaptation (QPA) (Liu et al., 2025) uses Quantum Neural Networks with hardware-efficient *ansätze* to predict weight parameters for LoRA adapter modules. We discuss these PEFT methods further in Section 3.3 and Appendices A A.2.2.

## 2.2 HAMMING-WEIGHT PRESERVING QUANTUM COMPUTING

As we will discuss, the generality of QuanTA (Chen et al., 2024) tensors, and the barren plateau features of hardware-efficient ansätze used in QPA (Liu et al., 2025) are problematic features for quantum computer deployment. On the other hand, subspace preserving quantum machine learning (QML) models have gained traction in the QML literature for their interpretability, analogies to classical counterparts and favorable training properties (Cherrat et al., 2023; Fontana et al., 2023; Monbroussou et al., 2024; Landman et al., 2022). Some HW preserving quantum models include Vision Transformers (Cherrat et al., 2024), Convolutional (Monbroussou et al., 2025; Mathur et al., 2025), Orthogonal (Landman et al., 2022) Neural Networks and quantum Mixture of Experts (MoE) models (Coyle et al., 2024). They have found applications in finance (Cherrat et al., 2023; Ramos-Calderer et al., 2021; Thakkar et al., 2024), medical imaging (Landman et al., 2022) and clinical data analysis (Kazdaghli et al., 2023). We will use these methods to construct quantum-inspired versions, and show their use in finetuning large foundation models. We include further technical details for these operations in Appendix D.

## 3    QUANTUM-INSPIRED COMPOUND ADAPTERS

In this section, we introduce Compound operations, the core of QuIC adapters, which leverage Hamming-weight preserving quantum circuits discussed in the previous section and can implement orthogonal and compound transformations on data. Inspired by these principles, we propose to construct quantum-inspired adapters using compound matrices up to a certain maximum Hamming-weight $K$. Combining compounding with orthogonality allows us to create novel adapters which are both expressive and parameter-efficient.

### 3.1    COMPOUND MATRICES

Given a 'base' matrix, $A \in \mathbb{R}^{n \times n}$, the *compound* matrix, $\mathcal{C}_k := A^{(k)}$, of 'order' $k \in [n]$ is defined as the $\binom{n}{k} \times \binom{n}{k}$ dimensional matrix with entries $A_{IJ}^{(k)} := \det(A_{IJ})$ where $I$ and $J$ are subsets of rows and columns of $A$ with size $k$. We use $\mathcal{C}_k$ as compact notation for our experiments later in the text. The work of (Kerenidis & Prakash, 2022) demonstrated how the action of these matrices on different Hamming-weight (different orders, $k$) could be efficiently performed using quantum circuits composed of so-called *fermionic beam splitter* (FBS) quantum gates. We will describe the quantum implementation in further detail later in the text.

However, we say that the *Compound Adapters* which serve the basis of our proposal are Quantum-*Inspired* because, for a constant Hamming-weight $k = \mathcal{O}(1)$, the action of these layers can be efficiently classically simulated by direct simulation of the subspaces. We will primarily deal with small order (and combinations thereof) compound matrices in this work, though we leave the open possibility of quantum speedups by quantum implementation of compound layers (Cherrat et al., 2023) to future work.

### 3.2    QUANTUM-INSPIRED COMPOUND ADAPTERS

Given a pre-trained weight matrix $W^* \in \mathbb{R}^{d \times d}$, we aim to construct a quantum-inspired adapter $\Delta W_Q \in \mathbb{R}^{d \times d}$ such that $W_{\text{adapt}} = \Delta W_Q W^*$. Now, the Quantum-Inspired Compound (QuIC) Adapter $\Delta W_Q$ is constructed using nested blocks, $\{\Delta W_Q^i\}_{i=1}^N$, each of which built via direct sum of compound matrices up to chosen order $K$, $\{A^{(k)}\}_{k=1}^K$:

$$\Delta W_Q = \bigoplus_{i=1}^N \Delta W_Q^i, \qquad \Delta W_Q^i := \begin{bmatrix} \Delta W_Q^{i,*} & 0 \\ 0 & \mathbb{1}_{b-d_{\text{comp}}} \end{bmatrix}, \qquad \Delta W_Q^{i,*} := \bigoplus_{k=1}^K A_i^{(k)}, \qquad (1)$$

where $d_{\text{comp}} := \sum_{k=1}^K \binom{n}{k}$. Each block is square $\Delta W_Q^i \in \mathbb{R}^{b \times b}, \forall i$ and $\bigoplus$ denotes the direct sum, i.e. $X \oplus Y := \begin{bmatrix} X & 0 \\ 0 & Y \end{bmatrix}$. This decomposition, similarly to OFT, introduces a 'block-size' hyperparameter, $b := {}^d/_N$, to regulate the total number of parameters. Therefore, each block is written explicitly as the block diagonal:

$$\Delta W_Q^i := \text{diag}(A_i^{(1)}, A_i^{(2)}, \mathbb{1}_{i,b-d_{\text{comp}}}) \qquad (2)$$

We show some examples of possible configurations in Figure. 3. Notably, when using only the first-order compound ($\mathcal{C}_1$), QuIC reduces to the OFT, demonstrating that our framework encompasses existing methods as special cases.

**Orthogonality:**    Compound matrices, $A^{(k)}$, inherit many properties from their base, $A$. These include for example, invertibility, positive definiteness and, importantly for us, unitarity and orthogonality. By constructing adapter blocks $\Delta W_Q^i$ using orthogonal compounds and padding with identities, orthogonality is preserved and inherited by $\Delta W_Q$. We test the importance of orthogonality as a property for our compound adapters later in section 5. This orthogonality preservation is formalized through the following Lemma (proof given in Appendix B):

**Lemma 1** (Orthogonality preservation of compound matrices). *If a base matrix, $A \in \mathbb{R}^{n \times n}$ is orthogonal, then all compound matrices, $A^{(k)}$ with $k \in [n]$, are orthogonal (and hence all QuIC Adapters). Furthermore, this orthogonality is preserved during finetuning when constructed with Hamming-weight preserving operations.*

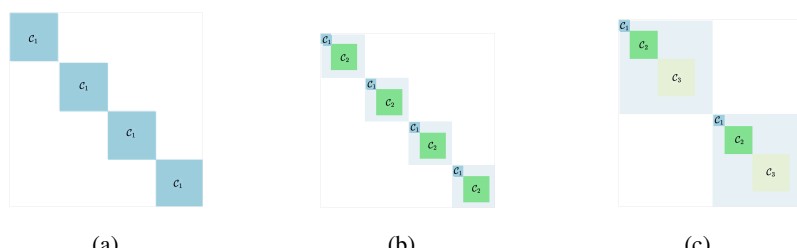

|     (a)     |     (b)     |     (c)     |

Figure 3: **Different possible QuIC Adapter configurations.** The adapter decomposition is determined by the number of blocks ($b$, or equivalently the 'rank' $r := d/b$), and the number of compounds within each block. Trailing dimensions are padded with an identity matrix, and are not trainable. The figure shows a) $\mathcal{C}_1$, $b = 4$ blocks, b) $\mathcal{C}_1 \oplus \mathcal{C}_2$, $b = 4$ blocks and c) $\mathcal{C}_1 \oplus \mathcal{C}_2 \oplus \mathcal{C}_3$, $b = 2$ blocks. Note, if the base matrix, $A$, is orthogonal then configuration (a) recovers OFT exactly.

**Parameter Efficiency:**   If the PLM matrix size is fixed, the number of trainable parameters is directly controlled by the tuple, $(b, K)$, the number of blocks and number of compounds therein. Choosing a larger $K$ reduces the possible size of the base matrix which can be compounded, $n$. All trainable adapter parameters are contained within this base matrix. This results in a compact parameterization suitable for large models. However, the compounding operation builds complex interactions between the parameters in higher orders. We show in our results that this is sufficient to gain high quality results with minimal tuning. The number of trainable parameters and complexity is given by the following Lemmata, which can be proved via simple parameter counting and in Appendix B

**Lemma 2** (Parameter Count of QuIC Adapters). *Let the PLM matrix $W \in \mathbb{R}^{d \times d}$ be partitioned into $N$ diagonal blocks, each of dimension $b := d/N$. For a maximum compound order $K$, the total number of non-zero entries is given by $P_{\text{non-zero}} = N \sum_{k=1}^{K} \binom{n}{k}^2 + \left( d - N \sum_k \binom{n}{k} \right)$ where $n = \max \left\{ m \in \mathbb{Z}_{>0} \middle| \sum_{k=1}^{K} \binom{m}{k} \leq b \right\}$. Moreover, the number of trainable parameters is given by $P_{\text{train}}^{\text{share}} = n^2$ if parameters are shared across blocks and $P_{\text{train}} = Nn^2$ if not. If orthogonality is enforced, we have $P_{\text{train}}^{\text{orth,share}} = \frac{1}{2}n(n-1), P_{\text{train}}^{\text{orth}} = \frac{1}{2}Nn(n-1)$.*

**Lemma 3** (Computational Complexity of QuIC Adapters). *Let a QuIC adapter $\Delta W_Q$ be defined for a layer of dimension $d$ with $N$ blocks, from a base matrix of size $n \times n$ and maximum compound order $K$.*

1. *The complexity of the forward pass (applying $\Delta W_Q$) is $\mathcal{O}(d^2/N)$.*

2. *The construction of $\Delta W_Q$ is a one-time cost, polynomial in $n$ for constant $K$. If parameters are shared, this cost is incurred once per layer.*

**Necessity of Combinatorial Compression with Determinants:**   One might ask whether the parameter efficiency is simply the result of expanding the effect of a small matrix into a combinatorially large space, and whether taking the determinant on minors could be replaced by another operation. We test this hypothesis by replacing the determinant with maximum and averaging operations. For instance, instead of constructing $A^{(k)}$ via $A_{IJ}^{(k,\texttt{comp})} := \det(A_{IJ})$ we test the following two element-wise on the matrix minors, $A_{IJ}^{(k,\texttt{max})} = \max(A_{IJ})$, i.e. taking the maximum element over minors, and $A_{IJ}^{(k,\texttt{avg})} = \text{avg}(A_{IJ})$, i.e. averaging over them. We find both of these operations perform poorly compared to the determinant, possibly because they do not respect orthogonality for multiplicative adapters. The determinant operation creates complex parameter interactions that enable extreme compression while preserving model expressiveness. We leave open the possibility that they may yet be performant alternatives for compound versions of additive adapters (e.g. LoRA).

### 3.3   QUANTUM NATIVE FINETUNING

Our primary proposal in this work is quantum-*inspired* finetuning, however here we briefly discuss quantum-*native* finetuning, where a quantum computer is actually used within the pipeline, either to

perform faster inference, or to continue finetuning with more expressive models. We expand on this discussion and detail relevant terminology in Appendix D. Importantly, as the maximum compound order $(K)$ increases, the compound circuits from which we derive inspiration become more difficult to classically simulate, increasing the potential for a speedup (even polynomial) when implemented quantum-natively.

As alluded to above, alternative Quantum-Inspired PEFT methods such as QuanTA (Chen et al., 2024) do not possess this native translation ability. A main motivation of QuanTA is the natural synergy between TNs and quantum circuits - much like our QuIC Adapters - ultimately with the potential of performing finetuning directly on quantum computers, perhaps using the computationally limited tensor networks for pre-training (Dborin et al., 2022; Rudolph et al., 2023). The QuanTA tensors, however, if scaled to large bond dimensions and qubit numbers $(n)$ require efficient (meaning polynomial in $n$) unitary compilation schemes for a quantum implementation (Dborin et al., 2022; Rudolph et al., 2023), which do not exist in general (Shende et al., 2006). Secondly, approaches such as QPA (Liu et al., 2025) also have the potential for quantum-native fine tuning but suffers from prohibitive measurement costs. QPA takes $2^N$ output probabilities from trainable quantum circuits on $N$ qubits, and maps to $M$ parameters (via a post-processing MLP) in a PEFT adapter (e.g. LoRA weight matrices). As such, only $N = \mathcal{O}(\log_2(M))$ qubits are required in the quantum circuit as an e.g. 30 qubit system has $2^{30} \approx 1B$ possible outcomes. However, to actually implement QPA as proposed on quantum hardware for $M = 1B$ parameters would necessitate $\mathcal{O}(2^N/\varepsilon^2) \approx 10,000$ billion measurement shots, accounting for $\varepsilon = 0.01$-accurate tomography. We discuss this further in Appendix A.2.2.

In contrast, for QuIC Adapters, we have a native classical-quantum translation, using similar concepts from recent proposals for Quantum Orthogonal Neural Networks (Landman et al., 2022). This translation arises because one only needs to train the parameters of the Hamming-weight preserving RBS/FBS gates rather than the parameters in their matrix representation. As such, the trained operation is always "compiled", and ready for quantum deployment. Direct readout of the final states is proportional to the maximum HW which is chosen, however alternative readout schemes can be designed for these circuits which retain much more efficiency (Cherrat et al., 2023), but yet retain novel features from the quantum implementation.

Table 1: Results on the GLUE development set, finetuning the pre-trained DeBERTaV3-base model. # Params denotes the number of trainable parameters. Our method is evaluated with the best configuration, $\mathcal{C}' = \mathcal{C}_1 \oplus \mathcal{C}_2$, where orthogonality is enforced ($\gamma = 0$), parameter sharing across blocks is disabled ($\beta = 0$), and the number of blocks is set to $b = 3$. Memory denotes the memory required to store trained weights. Pareto = (Accuracy - 44.01) / $\log_{10}$(params in K), where 44.01% is the DeBERTa-V3-base zero-shot mean. Frontier indicates methods on the Pareto-optimal curve.

| Method | # Params | SST-2 | CoLA | RTE | MRPC | STS-B | All | Memory (MB) | Pareto (↑) | Frontier |
|---|---|---|---|---|---|---|---|---|---|---|
| Full Finetuning | 184M | 95.63 | 69.19 | 83.75 | 89.46 | 91.60 | 85.93 | 702.0 | 7.96 | × |
| LoRA$_{r=8}$ (Hu et al., 2021) | 1.33M | 94.95 | 69.82 | 85.20 | 89.95 | 91.60 | 86.30 | 5.3 | 13.54 | × |
| OFT$_{b=16}$ (Qiu et al., 2023) | 0.79M | 96.33 | **73.91** | 87.36 | 92.16 | 91.91 | 88.33 | 3.0 | 15.29 | × |
| BOFT$_{b=8}^{m=2}$ (Liu et al., 2023) | 0.75M | **96.44** | 72.95 | **88.81** | **92.40** | **91.92** | **88.50** | 2.9 | 15.47 | ✓ |
| DoRA (Liu et al., 2024) | 0.55M | 94.98 | 64.90 | 79.15 | 89.72 | 91.28 | 84.00 | 2.0 | 14.59 | × |
| AdaLoRA (Zhang et al., 2023) | 0.32M | 95.80 | 70.04 | 87.36 | 90.44 | 91.63 | 87.05 | 1.3 | 17.18 | ✓ |
| BitFit (Ben Zaken et al., 2022) | 0.1M | 94.84 | 66.96 | 78.70 | 87.75 | 91.35 | 83.92 | 0.4 | 19.95 | × |
| QuanTA$_{16-16-4-4}$ (Chen et al., 2024) | 0.093M | 95.30 | 67.75 | 84.48 | 89.22 | 91.01 | 85.55 | 0.4 | 21.10 | × |
| LoKr (YEH et al., 2024) | 0.073M | 95.07 | 69.46 | 85.20 | 89.71 | 90.76 | 86.04 | 0.3 | 22.56 | ✓ |
| QuIC$_{\mathcal{C}_1 \oplus \mathcal{C}_2}$ | 0.03M | 94.83 | 68.04 | 84.03 | 89.95 | 91.04 | 85.57 | **0.12** | **28.14** | ✓ |

# 4 EXPERIMENTAL SETUP

## 4.1 MODEL AND DATA

We evaluate the effectiveness of our QuIC Adapters by finetuning multiple moderate sized and large foundation models on a comprehensive selection of datasets over several areas. In particular, our experiments span four distinct domains, natural language understanding, computer vision, discrete reasoning and math. For language understanding, we use the GLUE benchmark (Wang, 2018). For the computer vision application, we incorporate the Visual Task Adaptation Benchmark (VTAB). For math problems, we use the MATH10K (Hu et al., 2023) and for reasoning, we use the Discrete

Reasoning Over the text in the Paragraph (DROP) dataset (Dua et al., 2019), which is an English reading comprehension benchmark requiring both natural language understanding and discrete reasoning operations.

We utilize the pre-trained DeBERTaV3-base model (He et al., 2021) as the backbone for our natural language experiments. For vision tasks, we employ the pre-trained DINOv2-large model (Oquab et al., 2023) as our backbone. Finally, for a larger scale model, we finetune LLaMA 7B (Touvron et al., 2023a) on math and discrete reasoning tasks.

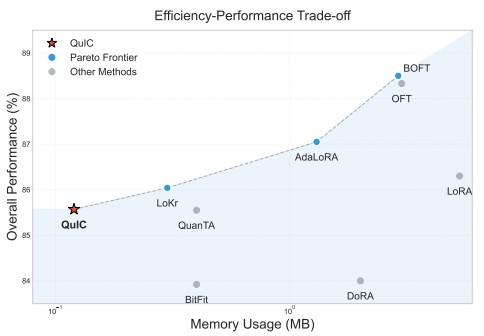 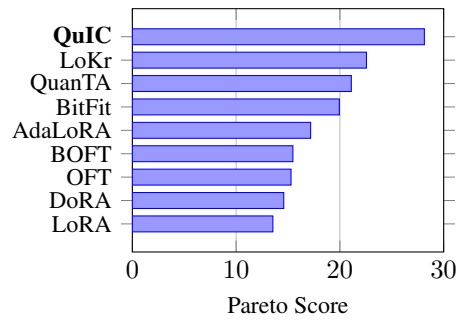

(a) Pareto frontier analysis showing aggregate GLUE accuracy versus log of trainable parameters. QuIC achieves optimal trade-off alongside BOFT, AdaLoRA, and LoKr.

(b) Pareto score for each PEFT model, averaged over GLUE. QuIC achieves the highest efficiency.

Figure 4: Performance analysis of QuIC and baseline PEFT methods on GLUE benchmark

## 4.2 ADAPTER CONFIGURATIONS

We uniquely characterize a QuIC Adapter configuration by a tuple $(\mathcal{C}', O, b, \gamma, \beta)$ given a maximum possible compound order (Hamming-weight), $K$. $\mathcal{C}'$ is the collection of Compounds used to construct the direct sum, e.g. $\mathcal{C}' = \mathcal{C}_1$ (including only the base matrix) or $\mathcal{C}' = \mathcal{C}_1 \oplus \mathcal{C}_2 \oplus \mathcal{C}_3$ (including compound matrices up to order 3. $\mathcal{O} \in \{\text{comp}, \text{max}, \text{avg}\}$ is the operation used to construct combinatorial operations. The final parameters $b, \gamma, \beta$ determine the block size, whether orthogonality is used and whether parameter sharing across blocks is applied, respectively.

## 5 RESULTS AND ANALYSIS

Our experiments demonstrate the effectiveness of QuIC adapters in achieving significant parameter efficiency while maintaining competitive performance across various GLUE benchmark tasks. In this section, we present an analysis of the trade-offs between parameter count and model accuracy, the combined impact of orthogonality and component-wise performance differences.

**QuIC Adapters for Language** We begin by finetuning on the GLUE (Wang, 2018) language benchmark with the state of the art PEFT methods in Table 1. GLUE encompasses a variety of natural language understanding tasks such as CoLA for grammatical acceptability (Warstadt, 2019), SST-2 for sentiment analysis (Socher et al., 2013), MRPC (Dolan & Brockett, 2005) and RTE (Dagan et al., 2006) for textual entailment, and STS-B (Cer et al., 2017) for semantic similarity. We use the best QuIC configuration found, which is $(\mathcal{C}_1 \oplus \mathcal{C}_2, \text{comp}, b = 3, \gamma = 0, \beta = 0)$, in other words - enforcing orthogonality without block-share over $b = 3$ blocks.

It can be seen from the Table (also seen with other datasets below) that QuIC Adapters do not generally outperform other methods in terms of raw accuracy or score. However, they are clearly far more performant relative to available parameter counts, and memory required to store weights. To formalize this, we use the Pareto score - defined as (Accuracy - baseline) / $\log_{10}(\text{params in K})$, which measures the efficiency-accuracy trade-off. From the Table and Figure 4a, QuIC Adapters achieve a state-of-the-art Pareto score of 28.14, placing them on the Pareto frontier alongside BOFT, AdaLoRA, and LoKr.

**QuIC Adapters for Vision** Next, for the computer vision application, we incorporate the Visual Task Adaptation Benchmark (VTAB), selecting datasets across *natural* images, *specialized* remote sensing imagery, and *structured* 3D environments. Specifically, CIFAR-100 (Krizhevsky, 2009), Pets (Parkhi et al., 2012), and nat-

Table 2: Results on a subset of the VTAB1k benchmark, finetuning the pre-trained DINOv2-large model. # Params denotes the number of trainable parameters. For the QuIC Adapter, we use the configuration, $(\mathcal{C}_1 \oplus \mathcal{C}_2, b = 3, \gamma = 0, \beta = 0)$.

| Method | # Params (M) | CIFAR100 | Pets | SVHN | Resisc45 | DMLab | Avg | Pareto (↑) |
|---|---|---|---|---|---|---|---|---|
| Full Finetuning | 304.4 | 67.6 | 93.7 | 92.8 | 90.9 | 58.1 | 80.62 | 0.26 |
| LoRA$_{r=4}$ | 1.77 | 77.2 | 94.8 | **94.7** | 91.4 | 58.1 | 83.24 | 47.01 |
| OFT$_{b=16}$ | 2.10 | 77.7 | 94.7 | 92.9 | 91.5 | 60.5 | 83.46 | 39.74 |
| BOFT$_{m=2,b=8}$ | 1.99 | 78.1 | **95.0** | 93.0 | **91.6** | **61.4** | **83.82** | 42.11 |
| QuIC$_{\mathcal{C}_1 \oplus \mathcal{C}_2}$ | 0.13 | **87.5** | 94.04 | 89.98 | 88.79 | 54.74 | 82.61 | **635.46** |

ural images, focusing on general object classification, fine-grained pet breed identification, and digit recognition from real-world street numbers, respectively. For a specialized dataset, RESISC45 (Cheng et al., 2017) contains remote sensing imagery - evaluating models on aerial scene classification. Finally, DMLab (Beattie et al., 2016) is an example of a structured dataset, derived from 3D navigation and interactive environments, testing visual reasoning through agent-based observations.

Table 3: Results on (a) a math benchmark (MATH10K) and (b) a discrete reasoning task (DROP), finetuning LLaMA 7B. We use the configuration $(\mathcal{C}_1 \oplus \mathcal{C}_2, b = 4, \gamma = 0, \beta = 0)$ for all cases.

(a)

| Method | # Params | AQUA | GSM8K | MAWPS | SVAMP | Avg | Pareto (↑) |
|---|---|---|---|---|---|---|---|
| Full FT | 7B | 19.3 | 65.2 | 92.0 | 80.7 | 64.3 | 0.009 |
| LoRA$_{r=32}$ | 58.1M | 17.5 | 65.7 | 91.2 | **80.8** | **65.6** | 1.12 |
| QuanTA$_{16-16-4-4}$ | 13.3M | 16.7 | **67.0** | **94.3** | 80.3 | 64.5 | 4.85 |
| QuIC$_{\mathcal{C}_1 \oplus \mathcal{C}_2}$ | 0.5M | **24.8** | 45.9 | 69.3 | 69.9 | 52.1 | **104.2** |

(b)

| Method | # Params | DROP | Pareto (↑) |
|---|---|---|---|
| Full FT | 7B | 59.4 | 0.008 |
| LoRA$_{r=32}$ | 17.5M | 54.0 | 3.08 |
| QuanTA$_{16-16-4-4}$ | 13.3M | **59.5** | 4.47 |
| QuIC$_{\mathcal{C}_1 \oplus \mathcal{C}_2}$ | 0.5M | 52.6 | **105.2** |

We also reduce the number of examples in each dataset to create VTAB1k (Zhai et al., 2019) where 1000 random labeled datapoints are used for training and validation, but the final accuracies we show are computed on the entire original VTAB test dataset. We use the same QuIC configuration as with GLUE. Here, we observe QuIC Adapters achieve superior Pareto scores, demonstrating excellent efficiency-accuracy trade-offs in vision tasks. Interestingly, in contrast with the other datasets across vision and NLP we test, CIFRAR100 stands out as having significantly *increased* accuracy relative to other methods, on the order of 10%.

**QuIC Adapters for Math** Next, we test the ability of QuIC Adapters to scale to larger models. To do so, we finetune LLaMA-2 7B (Touvron et al., 2023a), a 7 billion parameter model released by Meta AI. We use a subset of the MATH10K dataset which is a multi-task arithmetic reasoning corpus introduced by Hu et al. (Hu et al., 2023), and use four of its established math word-problem benchmarks: Grade School Math 8K (GSM8K), Simple Variations on Arithmetic Math word Problems (SVAMP), MAth Word ProblemS (MAWPS), and Algebra Question Answering with Rationales (AQuA).

**QuIC Adapters for Reasoning** Finally, we test QuIC Adapters on a discrete reasoning task, using LLaMA-7B and finetuning it over the Discrete Reasoning Over the text in the Paragraph (DROP) dataset (Dua et al., 2019). It is a benchmark designed to evaluate language models' advanced reasoning capabilities through complex question answering tasks. It encompasses over 9500 intricate challenges that demand numerical manipulations, multi-step reasoning, and the interpretation of text-based data.

## 6 ABLATION STUDIES ON GLUE

**Increasing parameters:** From Table 4 we can see two features of our adapters. First, the hyper compression offered by the combinatorial compounding operation, does not allow a large flexibility in changing the number of trainable parameters. Once a non-trivial compound matrix has been added to the adapter (i.e. of greater order than compound 1), the parameter count reduces dramatically. To address this, we can increase the parameter count monotonically by *multiplying* several QuIC Adapters. This is a general concept applicable to both additive or multiplicative adapters. For example,

in Table 5 we demonstrate that using 4 multplicative adapters can improve the performance across all GLUE datasets, still without dramatically increasing the parameter count.

Table 4: Summary of configurations with their respective parameter counts and accuracies on the STS-B dataset with the best configurations in bold. If the base matrix is $A$ then $A^{(k)} =: \mathcal{C}_k$. Increasing maximum compound order, $K$, necessitates a reduction in trainable parameters.

| Configuration | Base matrix | Params | Accuracy (%) | Configuration | Base matrix | Params | Accuracy (%) |
|---|---|---|---|---|---|---|---|
| $\mathcal{C}_1 \equiv$ OFT | $A \in \mathbb{R}^{256 \times 256}$ | 1,770,241 | **91.68** | $\mathcal{C}_1 \oplus \mathcal{C}_2$ | $A \in \mathbb{R}^{22 \times 22}$ | 33,217 | **88.85** |
| $\mathcal{C}_2$ | $A \in \mathbb{R}^{23 \times 23}$ | 38,401 | 40.57 | $\mathcal{C}_1 \oplus \mathcal{C}_3$ | $A \in \mathbb{R}^{12 \times 12}$ | 16,321 | **88.53** |
| $\mathcal{C}_3$ | $A \in \mathbb{R}^{12 \times 12}$ | 16,321 | 42.20 | $\mathcal{C}_2 \oplus \mathcal{C}_3$ | $A \in \mathbb{R}^{11 \times 11}$ | 13,057 | 40.60 |
| | | | | $\mathcal{C}_1 \oplus \mathcal{C}_2 \oplus \mathcal{C}_3$ | $A \in \mathbb{R}^{11 \times 11}$ | 13,057 | **88.48** |

The second observation from Table 4 is the first part of our ablation study. It is clear from these results that the inclusion of the first order compound - the base matrix, $A =: \mathcal{C}_1$, is crucial to the success of QuIC Adapters. We hypothesize this is due to the difficulty of gradient flow through the determinant operation to the parameters in $A$, when $A$ itself is not included.

**Impact of orthogonality:** The second ablation study we conduct is the impact of orthogonality on the QuIC Adapters (detailed results in Appendix D.4.2). Like the inclusion of $\mathcal{C}_1$, we also find including orthogonality is critical for QuIC Adapters. Focusing on STS-B, we find that adapter configurations *with* orthogonality can achieve a score of 68.70 when averaged over the configurations in Table 4, while non-orthogonal configurations achieve only an average of 27.32. The possible reason for this is the preservation of orthogonality by determinants, which is reinforced when we replace the determinant computation on minors with other combinatorial operations, such as `max` and `avg`

Table 5: Increasing parameter count in QuIC Adapters. Trainable parameter count can be naturally increased by multiplying successive adapters, leading to performance boosts. Here we compare a single adapter, $\Delta W_Q$ versus four, $\prod_{\ell=1}^{4} (\Delta W_Q^\ell)$.

| Method | # Params | CoLA | RTE | MRPC | STS-B |
|---|---|---|---|---|---|
| Full Finetuning | 184M | 69.19 | 83.75 | 89.46 | 91.60 |
| LoRA$_{r=8}$ | 1.33M | 69.82 | 85.20 | 89.95 | 91.60 |
| OFT$_{b=16}$ | 0.79M | 73.91 | 87.36 | 92.16 | 91.91 |
| BOFT$_{m=2,b=8}$ | 0.75M | 72.95 | 88.81 | 92.40 | 91.92 |
| QuIC$_{\mathcal{C}_1 \oplus \mathcal{C}_2}$ | 0.03M | 64.57 | 81.22 | 87.99 | 90.16 |
| QuIC$_{4 \times (\mathcal{C}_1 \oplus \mathcal{C}_2)}$ | 0.14M | 65.83 | 80.50 | 86.27 | 91.44 |

(we conduct this ablation study in Appendix D.4.3). Even poorly performing compound configurations, such as those without $\mathcal{C}_1$, see a significant performance boost when orthogonality is enforced. Finally, we note we show only the impact of orthogonality for *multiplicative* adapters. One could also consider QuIC Adapters in an additive form (similar to LoRA), which we leave to future work.

# 7 CONCLUSION

This work presents a novel proposal for parameter-efficient finetuning, leveraging quantum-inspired principles to construct efficient adapters with minimal additional parameters. Our results indicate that compound operation based adapters can serve as a promising alternative to existing PEFT methods (encompassing them in some cases), achieving substantial parameter reduction while maintaining strong performance across a range of language and vision tasks.

Our experiments reveal that against other quantum inspired peft techniques, QuIC adapters offer competitive performance while having a much better performance over parameter count budget. Furthermore, QuIC's natural translation ability on quantum hardware sets it apart from its counterparts and underscores its potential for broader applications in the future.

Future work will explore extending these ideas to more complex architectures, further optimizing adapter design, and investigating potential quantum adapter implementations. By bridging quantum-inspired techniques with deep learning, we hope to advance the field of efficient finetuning and enable scalable adaptation of large foundation models in practical settings.

## 8  Reproducibility Statement

We have taken several steps to ensure reproducibility of our results. The key hyperparameters, training settings, and evaluation protocols are reported in the Appendix (Section E). All datasets used are standard public benchmarks referenced appropriately in the main text and appendix. Finally, the full source code and instructions to reproduce our experiments are available in our anonymized repository: `https://anonymous.4open.science/r/quic-adapters-D41E/README.md`.

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

## A  EXTENDED BACKGROUND

In this section we provide more verbosity on the background and alternative finetuning methods discussed in the main text.

### A.1  TRANSFORMER ARCHITECTURE

The transformer architecture has become the foundation for many large language and vision foundation models due to its ability to capture long-range dependencies and its scalability. It consists of stacked encoder and decoder layers, each containing multi-head self-attention and feed-forward network layers. These components are interconnected by residual connections and layer normalization. PEFT methods typically focus on modifying the self-attention and feed-forward network (FFN) layers to introduce trainable parameters efficiently. We describe these layers briefly as follows:

**Multi-Head Self-Attention Layer:**  For an input sequence $X \in \mathbb{R}^{n \times d}$, where $n, d$ are the sequence length and hidden dimension respectively, the self-attention mechanism computes as follows: $\text{Attn}(Q, K, V) = \text{softmax}\left(QK^\top / \sqrt{d}\right) V$, where the query, key and value matrices, $Q = XW_Q$, $K = XW_K$, and $V = XW_V$ are linear projections of the input $X$ using learnable weight matrices $W_Q, W_K, W_V \in \mathbb{R}^{d \times d}$ respectively.

**Feed-Forward Network (FFN) Layer:**  A typical FFN layer involves two trainable weight matrices, $W_1 \in \mathbb{R}^{d \times d_F}$, $W_2 \in \mathbb{R}^{d_F \times d}$, and is defined as $\text{FFN}(X) = \sigma(XW_1 + \mathbf{b}_1)W_2 + \mathbf{b}_2$, where $d_F$ is the dimension of the feed-forward layer and $\sigma$ is a non-linear function which we assume to be $\sigma(\cdot) := \text{ReLU}(\cdot)$.

### A.2  ORTHOGONAL FINETUNING (OFT)

Orthogonal Finetuning (OFT) (Qiu et al., 2023) is an alternative approach to parameter-efficient finetuning which enforces an *orthogonality* constraint on the adapter. The authors justify orthogonality as a useful feature in helping preserve the hyperspherical energy i.e. the angular feature difference between neurons (Liu et al., 2018) which in turn helps preserve original knowledge of the model. Unlike methods such as LoRA that inject low-rank updates in an *additive* manner, OFT and its variants introduce *multiplicative* adapters. In this case, the updated weight matrix is expressed as:

$$W_{\text{OFT}} = \Delta W_{\text{OFT}} W^*, \tag{3}$$

Again, OFT assumes $W^* \in \mathbb{R}^{d \times d}$ is a square pre-trained weight matrix and $\Delta W_{\text{OFT}} \in \mathbb{R}^{d \times d}$ is the orthogonal adapter, where we have $\Delta W_{\text{OFT}}^\top \Delta W_{\text{OFT}} = \mathbb{1}$. The orthogonality of $\Delta W_{\text{OFT}}$ ensures that the transformation preserves the spectral properties of $W^*$, retaining the pre-trained knowledge during finetuning. Different parameterizations of $\Delta W_{\text{OFT}}$ are possible - specifically, (Qiu et al., 2023) chooses to employ the Cayley transform. Given a parameterized matrix, $P \in \mathbb{R}^{d \times d}$, the OFT adapter with the Cayley transform is defined as:

$$\Delta W_{\text{OFT}}^{\text{C}} := (\mathbb{1}_d + Q)(\mathbb{1}_d - Q)^{-1}, \quad Q := \frac{1}{2}(P - P^T) \tag{4}$$

The Cayley transform is efficient and ensures that $\Delta W_{\text{OFT}} \in \text{SO}(d)$, the special orthogonal group of dimension $d$. To further improve parameter efficiency, OFT introduces a block-diagonal structure to

$\Delta W_{\text{OFT}}$. The orthogonal matrix is partitioned into $r$ smaller orthogonal blocks, each parameterized with (4):

$$\Delta W_{\text{OFT}}^{\text{BD},r} := \begin{bmatrix} \Delta W_{\text{OFT},1}^{\text{C}} & 0 & \cdots & 0 \\ 0 & \Delta W_{\text{OFT},2}^{\text{C}} & \cdots & 0 \\ \vdots & \vdots & \ddots & \vdots \\ 0 & 0 & \cdots & \Delta W_{\text{OFT},r}^{\text{C}} \end{bmatrix} \tag{5}$$

where each $\Delta W_{\text{OFT},i} \in \mathbb{R}^{d/r \times d/r}$ and $Q_i \in \mathbb{R}^{d/r \times d/r}$. When $r = 1$, the block-diagonal matrix reduces to the original full orthogonal matrix, $\Delta W_{\text{OFT}}^{\text{BD},1} = \Delta W_{\text{OFT}}$. For the remainder of the text, we implicitly assume this block-diagonal structure in OFT and drop the superscripts when clear from context. Using this block-diagonal structure, the total number of parameters is reduced to $\mathcal{O}(d^2/r)$, which can be compressed further to $\mathcal{O}(d^2/r^2)$ via parameter sharing across blocks.

### A.2.1 BUTTERFLY ORTHOGONAL FINE-TUNING (BOFT)

As discussed briefly in the main text, Butterfly Orthogonal Fine-Tuning (BOFT) (Liu et al., 2023) extends OFT by introducing an efficient parameterization of the orthogonal matrix using butterfly structures. In BOFT, the orthogonal matrix $\Delta W_{\text{BOFT}} \in \mathbb{R}^{d \times d}$ is constructed as a product of $m$ sparse orthogonal matrices derived from '*butterfly*' structures:

$$\Delta W_{\text{BOFT}} = \prod_{i=1}^{m} \widetilde{B}_{(i)}, \tag{6}$$

where each $\widetilde{B}_{(i)} \in \mathbb{R}^{d \times d}$ is a butterfly *factor* - a sparse orthogonal matrix that efficiently captures global interactions within the data. These butterfly factors are recursively defined and constructed to ensure orthogonality. The butterfly structure originates from the Cooley-Tukey algorithm for the Fast Fourier Transform, known for its efficient information exchange properties. In BOFT, the butterfly factors are built using small orthogonal blocks that are combined to form larger orthogonal matrices. Specifically, each butterfly factor $\widetilde{B}_{(i)}$ is defined as, $\widetilde{B}_{(i)} = \text{Permute}\left(\text{diag}\left(\Delta W_{\text{BF},1}^{(i)}, \Delta W_{\text{BF},2}^{(i)}, \ldots, \Delta W_{\text{BF},k}^{(i)}\right)\right)$, where $\Delta W_{\text{BF},j}^{(i)} \in \mathbb{R}^{b \times b}$ are small orthogonal matrices parameterized via the Cayley transform (4), $k := d/b$ are the number of blocks at level $i$ and $\text{Permute}(\cdot)$ rearranges the blocks to create the butterfly pattern. They typically take the number of butterfly factors to be $m = \log_b d$ where $b$ is the block size, and $b \geq 2$. The number of parameters required is $N_P^{\text{BOFT}} = \frac{1}{2}md(b-1) = \frac{1}{2}(b-1)d\log_b d$ (Liu et al., 2023). When $b = 2$, the parameter count becomes $N_P^{\text{BOFT}} = \mathcal{O}(d\log d)$, compared to the $N_P^{\text{OFT}} = \mathcal{O}(d^2)$ parameters required for a full orthogonal matrix in OFT.

### A.2.2 QUANTA

Here, we give some extended background on Quantum-informed Tensor Adaptation (QuanTA) (Chen et al., 2024), an alternative Quantum-Inspired Adapter recently proposed.

Given the pre-trained matrix, $W \in \mathbb{R}^{d \times d}$, QuanTA constructs an adapter, $\Delta W_{\text{QuanTA}}$ as an additive adapter $W_{\text{adapt}} = W + \Delta W_{\text{QuanTA}}$[1]. The adapter, $\Delta W_{\text{QuanTA}}$ is constructed via *contraction* of multiple smaller tensors, first by factoring the original dimension input and output axes, $d, d$, into multiple (again smaller) tensorial axes $d \to \{d_1, d_2, \ldots, d_N\}$. Therefore, axis indexed by $n$ can be thought of as representing a $d_n$-dimensional quantum state (i.e. a qu$d_n$it). Most commonly, $d_n = 2, \forall n$, in which case the tensor adapter can be thought of as an operation on $N$ *qubits*.

Tensor networks are decompositions of tensors, i.e. the above QuanTA adapter, $\Delta W_{\text{QuanTA}} \in \mathbb{R}^{d_1, d_2, \ldots, d_N, d_1, d_2, \ldots, d_N}$ as a product of smaller tensors usually operating over fewer axes, e.g. three dimensional tensors, $\mathcal{T} \in \mathbb{R}^{d_r, d_s, d_t}$. The connected graph of $M$ of such tensors is called a tensor network. Tensor networks themselves have found use in machine learning applications for many years, with promising properties for developing and compressing machine learning models (Stoudenmire & Schwab, 2016; Novikov et al., 2015; Tomut et al., 2024).

---

[1]QuanTA also proposes an initialization strategy involving another contracted tensor network initialized to the same values as the adapter, but which remains frozen during training.

The full adapter is then constructed by *contracting* the network over all "virtual" or *bond* dimensions, and reshaping the "physical" dimensions (i.e. $\{d_i\}_{i=1}^N, \{d_j\}_{j=1}^N$) back to $d \times d$ for no-overhead inference. As given in (Chen et al., 2024), an $M = 3$ tensor example is:

$$\Delta W_{\text{QuanTA}} := \mathcal{T}, \mathcal{T}_{i;j} = \mathcal{T}_{i_1,i_2,i_3;j_1,j_2,j_3} = \sum_{k_1,k_2} \mathcal{T}^1_{i_1,i_2;k_1,k_2} \sum_{k_3} \mathcal{T}^2_{k_1,i_3;j_1,k_3} \mathcal{T}^3_{k_2,k_3;j_2,j_3} \tag{7}$$

In the above Eq. (7), each of $\mathcal{T}^1, \mathcal{T}^2, \mathcal{T}^3$ are 4 index tensors. Here, $\mathcal{T}^1/\mathcal{T}^2$ carries two/one physical input dimensions, $(i_1, i_2)/i_3$ respectively while $\mathcal{T}^2$ and $\mathcal{T}^3$ carry one/two physical output dimensions, $j_1/(j_2, j_3)$ respectively. All other dimensions $(k_1, k_2, k_3)$ are virtual/bond dimensions. Assuming the physical dimensions are fixed, the complexity of dealing with a tensor network contraction (multiplying over bond dimensions) is determined by the dimensions of the bond indices. This also directly regulates the number of trainable parameters within the model/adapter.

**Quantum circuit implementation:** Finally, if one wished to translate QuanTA tensors for further quantum-native finetuning (as we discuss in Appendix D) the means of doing so in general is still an open research question. Specifically, quantum computers require unitary operations, and at no stage in training will the tensors in QuanTA have unitarity enforced. Therefore, each of $\mathcal{T}^1, \mathcal{T}^2, \mathcal{T}^3$ will need to be canonicalised. The canonicalisation procedure makes each tensor an isometry via singular value decomposition through the network. The canonicalisation procedure also enables truncation of the network by clipping singular values. However if the resulting tensors are not square, they will need to be suitably constructed into a full unitary by some method.

Finally, assuming the tensors are not simply two-axes operators (two input and two output qubits), the resulting unitaries need to be compiled to the available gatesets of the quantum computer. One of the most efficient general purpose exact compilation schemes is via the Quantum Shannon Decomposition (QSD) which recursively compiles unitaries into smaller and smaller sub-blocks via de-multiplexing (Shende et al., 2006). The QSD requires $23/48 \times 4^n - 3/2 \times 2^n + 4/3$ CNOT gates to compile a general $2^n \times 2^n$ unitary over $n$ qubits, which is exponential in $n$.

# B   TECHNICAL PROOFS

Here, we give the proofs of the Lemmata from the main text.

**Lemma** (Orthogonality preservation of compound matrices (Lemma 1 repeated)). *If a base matrix, $A \in \mathbb{R}^{n \times n}$ is orthogonal, then all compound matrices, $A^{(k)}$ with $k \in [n]$, for are orthogonal. Furthermore, this orthogonality is preserved during finetuning if we maintain the orthogonality of the base matrix.*

*Proof.* Let $A \in \mathbb{R}^{n \times n}$ be an orthogonal matrix, i.e., $A^\top A = AA^\top = \mathbb{1}_n$. For any $k \in [n]$, the $k$-th compound matrix $A^{(k)}$ has entries $A_{IJ}^{(k)} := \det(A_{IJ})$ where $I$ and $J$ are $k$-element subsets of $[n]$. Now, to show that $A^{(k)}$ is orthogonal, we need to prove $(A^{(k)})^T A^{(k)} = I_{\binom{n}{k}}$.

Consider the $(I, J)$-entry of $(A^{(k)})^T A^{(k)}$:

$$(A_{IJ}^{(k)})^\top A_{IJ}^{(k)} = \sum_K A_{KI}^{(k)} \cdot A_{KJ}^{(k)} = \sum_K \det(A_{KI}) \cdot \det(A_{KJ}) \tag{8}$$

By the Cauchy-Binet formula, Eq. 8 equals $\det\big((A^\top A)_{IJ}\big)$. Since $A$ is orthogonal, we have $A^\top A = \mathbb{1}_n$, so:

$$\det\big((A^\top A)_{IJ}\big) = \det((I_n)_{IJ}) = \begin{cases} 1 & \text{if } I = J \\ 0 & \text{if } I \neq J \end{cases} \tag{9}$$

Therefore, $(A^{(k)})^\top A^{(k)} = \mathbb{1}_{\binom{n}{k}}$, proving that $A^{(k)}$ is orthogonal.

To maintain orthogonality during finetuning, we employ the Cayley parameterization as follows. We parameterize the base matrix $A$ using the Cayley transform: $A = (I + Q)(I - Q)^{-1}$ where $Q$ is a skew-symmetric matrix ($Q = -Q^\top$). During finetuning, we update only the entries of

$Q$ (maintaining its skew-symmetry), which automatically ensures that $A$ remains orthogonal with determinant 1 (i.e., $A \in \mathrm{SO}(n)$). The compound matrices $A^{(k)}$ are then computed directly from this orthogonal base matrix.

Alternatively, to preserve orthogonality during training, one could employ the quantum strategy of (Landman et al., 2022) described in Appendix D where the orthogonal/compound matrix is trained using its parameterization with Reconfigurable or Fermionic Beam Splitter RBS/FBS quantum gates.

$\square$

Here we provide a concrete example of how the dimensions of the QuIC adapter components are chosen to match the dimensionality of a pre-trained model's weight matrix. The primary constraint is that the sum of the dimensions of the compound matrices, $d_{\mathrm{comp}} = \sum_{k=1}^{K} \binom{n}{k}$, must be less than or equal to the block size, $b$. The base matrix dimension, $n$, is typically chosen to maximize this sum without exceeding $b$.

For example, consider a pre-trained weight matrix of size $d = 1024$, which we will adapt with a single block ($N = 1$, so $b = 1024$). If we choose a maximum Hamming-weight of $K = 2$, we need to find an integer $n$ such that $\binom{n}{1} + \binom{n}{2} \le 1024$. To maximize parameterization, we want the largest such $n$. The expression is $n + \frac{n(n-1)}{2} \le 1024$. A suitable choice is $n = 44$, which gives $d_{\mathrm{comp}} = 44 + \binom{44}{2} = 44 + 946 = 990$.

The identity matrix $\mathbb{1}_{b-d_{\mathrm{comp}}}$ is then added to pad the remaining $1024 - 990 = 34$ dimensions. With this example, the matrices in the block defined in Eq. 2 have the following dimensions: $A^{(1)} \in \mathbb{R}^{44 \times 44}$, $A^{(2)} \in \mathbb{R}^{946 \times 946}$, and the padding identity is $\mathbb{1}_{34} \in \mathbb{R}^{34 \times 34}$.

Alternatively, if we wished to maximize the *number* of compound orders for the same block size ($b = 1024$), we could choose $n = 11$ and $K = 5$. This would yield compound matrices $A^{(1)} \in \mathbb{R}^{11 \times 11}$, $A^{(2)} \in \mathbb{R}^{55 \times 55}$, $A^{(3)} \in \mathbb{R}^{165 \times 165}$, $A^{(4)} \in \mathbb{R}^{330 \times 330}$, and $A^{(5)} \in \mathbb{R}^{462 \times 462}$. The total dimension would be $d_{\mathrm{comp}} = 1023$, requiring only a single padding dimension ($\mathbb{1}_1 = 1$).

**Lemma** (Computational Complexity of QuIC Adapters (Lemma 3 repeated))**.** *Let a QuIC adapter $\Delta W_Q$ be defined for a layer of dimension $d$ with $N$ blocks, derived from a base matrix of size $n \times n$ and max compound order $K$.*

1. *The complexity of the forward pass (applying $\Delta W_Q$) is $\mathcal{O}(d^2/N)$.*

2. *The construction of $\Delta W_Q$ is a one-time cost, polynomial in $n$ for constant $K$. If parameters are shared, this cost is incurred once per layer.*

*Proof.* **1. Forward Pass Complexity:** The QuIC adapter $\Delta W_Q$ has a block-diagonal structure with $N$ blocks, each of size $b \times b$ where $b = d/N$. Applying this adapter to a vector involves $N$ independent multiplications with these smaller blocks. The cost for one block is $\mathcal{O}(b^2)$. The total cost is therefore:

$$N \times \mathcal{O}(b^2) = N \times \mathcal{O}\left(\left(\frac{d}{N}\right)^2\right) = N \times \mathcal{O}\left(\frac{d^2}{N^2}\right) = \mathcal{O}\left(\frac{d^2}{N}\right).$$

**2. Construction Complexity:** The construction of $\Delta W_Q$ from the base matrix $A \in \mathbb{R}^{n \times n}$ is dominated by generating the compound matrices $\{A^{(k)}\}_{k=1}^{K}$. To construct the $k$-th compound matrix, $A^{(k)}$, we compute the determinant of all $\binom{n}{k} \times \binom{n}{k}$ minors of size $k \times k$. The cost of a single $k \times k$ determinant is $\mathcal{O}(k^3)$. Thus, the total cost to construct $A^{(k)}$ is $\mathcal{O}(\binom{n}{k}^2 \cdot k^3)$.

The total construction cost sums over all compound orders up to $K$:

$$\mathrm{Cost} = \sum_{k=1}^{K} \mathcal{O}\left(\binom{n}{k}^2 \cdot k^3\right).$$

For a small, constant maximum order $K$, the complexity is dominated by the largest binomial coefficient term, where $\binom{n}{K} = \mathcal{O}(n^K)$. The total complexity is therefore $\mathcal{O}(n^{2K})$, which is polynomial in the base matrix size $n$. This construction cost is incurred only once per layer if parameters are shared across blocks, as the resulting matrices can be cached. $\square$

## C  ADAPTER CONFIGURATIONS (EXTENDED)

Here we elaborate on the different possible configurations of a QuIC Adapter. Our experimentation focused on different combinations of compound matrices based on Hamming-weights, the types of operations applied to these compounds, the enforcement of orthogonality, and the strategy for parameter sharing across adapter blocks.

Building upon this, we define compound matrices based on the Hamming-weight $k$ up to a maximum $K = 3$, constructed with $(I, J)$-minors such that $|I| = |J| = k$. We uniquely characterize an experiment by a tuple $(\mathcal{C}', O, b, \gamma, \beta)$. $\mathcal{C}'$ is a subset of all compound configurations (power set) constructed via direct sum, $\mathcal{C}' \subseteq \mathcal{P}(\mathcal{C})^{\oplus 3}$.

$$\mathcal{C} := \{\mathcal{C}_1, \mathcal{C}_2, \mathcal{C}_3\}, \mathcal{P}(\mathcal{C})^{\oplus 3} := \{\mathcal{C}_1, \mathcal{C}_2, \mathcal{C}_3, \mathcal{C}_1 \oplus \mathcal{C}_2, \mathcal{C}_2 \oplus \mathcal{C}_3, \mathcal{C}_2 \oplus \mathcal{C}_3, \mathcal{C}_1 \oplus \mathcal{C}_2 \oplus \mathcal{C}_3\} \qquad (10)$$

Note that this notation is slightly obfuscating. Given a fixed pre-trained matrix and block size, $d, b$, and two different configurations both containing the base matrix, e.g. $\mathcal{C}_1$ and $\mathcal{C}_1 \oplus \mathcal{C}_2 \oplus \mathcal{C}_3$. The base matrix compounded to construct the former configuration will be *larger* (and hence have more trainable parameters) than the one used to create the latter, in other words $\dim(A)_{\mathcal{C}_1} > \dim(A)_{\mathcal{C}_1 \oplus \mathcal{C}_2 \oplus \mathcal{C}_3}$ due to the dimension matching requirements. Therefore as the number of terms in the direct sum decreases along with the compound order, the number of trainable parameters is assumed to *increase*. One could of course restrict the definition $\mathcal{P}(\mathcal{C})^{\oplus}$ with a fixed base matrix size for all elements, and hence fixed number of parameters, but this may provide a bias in a different direction. As such, we keep the definition flexible and the implication of dimensions will be clear from context through the text.

Next, we have $O \in \{\texttt{comp}, \texttt{max}, \texttt{avg}\}$, defined as one of the dimensionality-expanding operations on minors from above, or 'compounding' - $\texttt{comp}$ - which refers to the usual determinant operation on minors. Orthogonality in the adapter matrices is regulated by the binary configuration parameter $\gamma \in \{0, 1\}$, with $\gamma = 0$ if orthogonality is enforced and $\gamma = 1$ otherwise. $\gamma = 0$ ensures the transformation preserves the norm and angles of the input feature vectors within the model.

Finally, $\beta \in \{0, 1\}$ is a block-share parameter - if $\beta = 1$, parameters are shared across adapter blocks and are distinct otherwise. A model with $\beta = 1$ will have fewer overall parameters than $\beta = 0$.

## D  QUANTUM IMPLEMENTATION

Our adapters, can be implemented efficiently on quantum hardware using fixed Hamming-weight encoders and Hamming-weight preserving circuits. Foremost among these are Hamming-weight (HW) preserving operations, which use quantum gates called Reconfigurable Beam Splitter (RBS) or their generalization into *Fermionic* Beam Splitter (FBS) gates. Circuits composed of these gates can be used on data encoded in states with a fixed (or multiple) Hamming-weight(s). As a specific example, take a vector $\mathbf{x} \in \mathbb{R}^{\binom{n}{2}}$. This vector can be *amplitude* encoded into the amplitudes of the state, restricted to those with Hamming-weight ($k = 2$). Specifically, we have $|\psi(\mathbf{x})\rangle := \frac{1}{\|\mathbf{x}\|} \sum_{e_k \in \mathrm{HW}_2^n} x_{e_k} |e_k\rangle$ where $e_k$ is a bitstring over $n$ (qu)bits with exactly 2 ones (and $n - 2$ zeros, e.g. $0101, 1010, 1001, 0011, 1100, 0110$ for $n = 4$). It turns out, that when circuits of FBS gates act on such states, their effective action on the vector is exactly that of the *compound* matrix of second-order, $\mathcal{C}_2 = A^{(2)}$ (Kerenidis & Prakash, 2022). In this section, we detail their implementation on quantum hardware.

### D.1  RECONFIGURABLE BEAM SPLITTER GATES

A Reconfigurable Beam Splitter $RBS$ gate is a two qubit gate parameterized with one angle $\theta \in [0, 2\pi]$. $RBS(\theta)_{ij}$ acting on the $i$-th and $j$-th qubits implements a Givens rotation:

$$\mathrm{RBS}_{ij}(\theta) = \begin{bmatrix} 1 & 0 & 0 & 0 \\ 0 & \cos(\theta) & \sin(\theta) & 0 \\ 0 & -\sin(\theta) & \cos(\theta) & 0 \\ 0 & 0 & 0 & 1 \end{bmatrix}$$

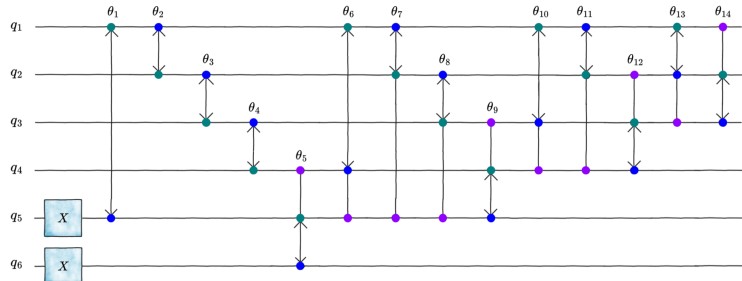

Figure 6: **A fixed Hamming-weight encoder.** Figure shows loading Hamming-weight-2 subspace ($k = 2$) in 6 qubits. Blue and green denote input and output respectively, violet denotes controlled operation. Figure from (Farias et al., 2024).

This is a Hamming-weight-preserving gate which is easy to implement on many quantum devices with compilations needing upto 2 CNOT gates with a pauli basis native gate set. Another Hamming-weight-preserving gate known as Fermionic Beam Splitter ($FBS$) gate which is a generalisation of $RBS$ gate could also be used to implement Hamming-weight-preserving circuits. The application of a $FBS$ between the qubits $i$ and $j$, $FBS_{ij}(\theta)$, acts as $RBS_{ij}(\theta)$ if the parity of the qubits between $i$ and $j$ is even, and is the conjugate gate $RBS_{i,j}(-\theta)$ otherwise. Therefore, in the case of unary inputs or nearest neighbour connectivity, $FBS$ and $RBS$ gates behave identically. The $FBS_{ij}$ is a non local gate that can be implemented using an RBS gate together with $\mathcal{O}(|i - j|)$ additional two qubit parity gates with a circuit of depth $\mathcal{O}(\log(|i - j|))$. We leave the discussion of quantum adapters using other Hamming-weight-preserving modalities like Linear Optics circuits for future work.

## D.2 Loaders

We shall use amplitude encoding to load classical data into the amplitudes of a quantum state. This involves mapping a data vector $x$ to a quantum state where the amplitudes of the basis states are proportional to the elements of $x$.

Unary encoding (Johri et al., 2021; Landman et al., 2022) is an amplitude encoding scheme that loads data into the amplitudes of computational basis states where each basis state has a Hamming-weight of 1. It uses $d$ qubits to encode a $d$-dimensional vector. Efficient quantum data encoders using $\mathcal{O}(d)$ two-qubit gates and $\mathcal{O}(\log d)$ depth are known in the unary basis as shown in Fig 5.

Fixed Hamming-weight (Hamming-weight-k) (Farias et al., 2024) encoding is an amplitude encoding scheme that loads a data vector into a subspace of fixed Hamming-weight $k$. It uses $n$ qubits to encode a data vector of size $d = \binom{n}{k}$, with $n \in \mathcal{O}(kd^{1/k})$. The circuit is constructed using a sequence of controlled (RBS) gates. The total CNOT-gate count for Hamming-weight-k encoding is $\mathcal{O}(kd)$. This type of encoding is an intermediate regime between unary and binary encodings.

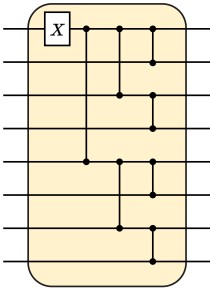

Figure 5: **A Unary loader.** Vertical lines denote parameterized RBS gates. Figure from (Cherrat et al., 2023). The input is $|0\rangle^{\otimes n}$ and the output is the loaded state in unary, $|\boldsymbol{x}\rangle = \frac{1}{||\boldsymbol{x}||} \sum_i x_i |e_i\rangle$, when read from left to right.

For our work, we require a quantum circuits capable of loading data vectors into subspaces of varying Hamming-weights, specifically from Hamming-weight 1 up to a maximum Hamming-weight k. This can be achieved by utilizing a series of fixed Hamming-weight (Hamming-weight-k) encoders, each dedicated to loading data into a subspace of a specific Hamming-weight. To load data up to

Hamming-weight k, we can sequentially stack the Hamming-weight-k encoders for each weight from 1 to k. The total number of qubits required is still n, but the total number of basis states becomes $\sum_{k=1}^{K} \binom{n}{k}$. This technique is distinct from a full binary encoder that includes all Hamming-weights from 0 to n. The overall CNOT gate count for such a construction can be expressed as the sum of CNOT gates for individual Hamming-weight-$k$ encoders, where $k$ varies from 1 to $K$, i.e.,

$$\text{Total CNOT count} = \sum_{k=1}^{K} \mathcal{O}\left(k\binom{n}{k}\right) \leq \mathcal{O}(d \log d), \text{ where } d = \binom{n}{K} \tag{11}$$

## D.3 LAYERS

Let $G(i, j, \theta)$ denote the Givens rotation applied to the $i$-th and $j$-th unary basis vector, i.e. $e_i$ and $e_j$, $\theta$ a vector of angles, and $\mathcal{T}$ is a list of triplets $(i, j, m)$. The Hamming-weight-preserving layer is defined by:

$$U(\theta) = \prod_{(i,j,m)\in\mathcal{T}} \text{RBS}_{ij}(\theta_m).$$

It acts as $U(\theta) |\mathbf{x}\rangle = W |\mathbf{x}\rangle$ where $W = \prod_{(i,j,m)\in\mathcal{T}} G(i, j, \theta_m)$.

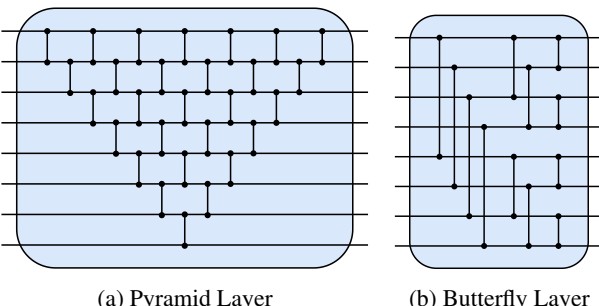

(a) Pyramid Layer          (b) Butterfly Layer

Figure 7: **Hamming-weight preserving layers.** Dots and dashes denote parameterised RBS gates. Figure from (Cherrat et al., 2023).

There are different circuits for $U(\theta)$, highlighted in Figure 7. The Pyramid architecture, as described in (Landman et al., 2022), consists of $n(n-1)/2$ RBS gates arranged in a pyramid-like structure and has a linear depth. This architecture allows for the representation of all possible orthogonal matrices of size $n \times n$. The Butterfly architecture, which was proposed in (Cherrat et al., 2024), in uses logarithmic depth circuits with a linear number of gates to implement a quantum orthogonal layer. This architecture, classical Cooley–Tukey algorithm used for Fast Fourier Transform, requires all-to-all connectivity in the hardware layout.

### D.3.1 QUANTUM IMPLEMENTATION

We can use these tools to construct quantum native implementation of our adapters as shown in figure 8. The block diagonal structure of our adapters imply that the adapters can be implemented via separate quantum circuits. For example in figure 8a, a 4 block $\mathcal{C}_1$ adapter can be implemented via 4 quantum circuits, each with Hamming-weight-1 loaders, a Hamming-weight-preserving layer and suitable measurements. Enforcing block share in this setting would imply the circuit layers sharing the same parameter values, however, the loaders still ought to be different. Similarly in figure 8b, we use 2 quantum circuits each with Hamming-weight-1, Hamming-weight-2 and Hamming-weight-3 loaders stacked one after another. Note that as specified in the binary encoders of (Farias et al., 2024), we would need parameterised $R_Y$ gates between each loader to enable sequential stacking.

## D.4 ABLATION STUDIES ON STS-B DATASET

To further understand the impact of different configuration setups, we run ablation studies on a dataset from the GLUE benchmark, specifically STS-B.

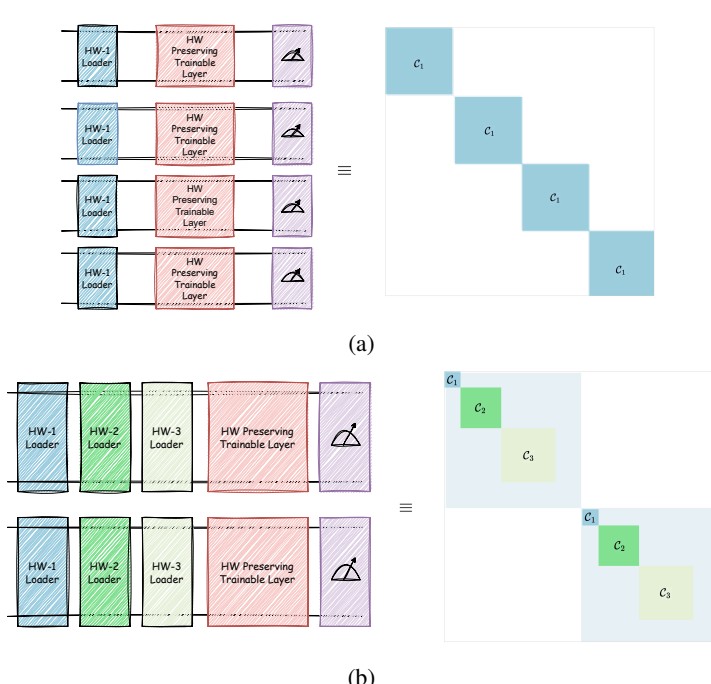

(a)

(b)

Figure 8: **Quantum Implementation of Adapters**. Each QuIC Adapter has an efficient quantum implementation using fixed Hamming-weight encoders and Hamming-weight preserving layers. Trailing dimensions are padded with an identity matrix. The figure shows quantum circuits for a) $\mathcal{C}_1$, $b = 4$ blocks, which uses only Hamming-weight 1 loaders and b) $\mathcal{C}_1 \oplus \mathcal{C}_2 \oplus \mathcal{C}_3$, $b = 2$ blocks which uses upto Hamming-weight 3 loaders.

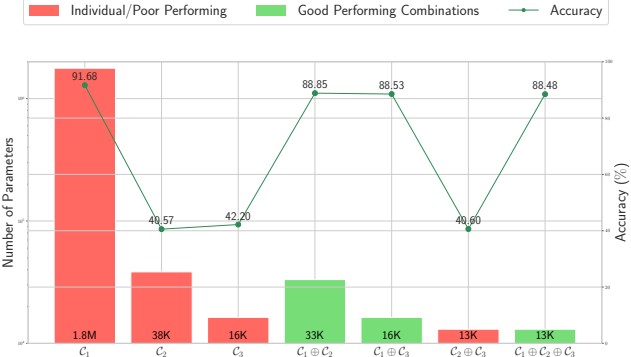

Figure 9: Visualization of performance versus parameter count for different adapter combinations.

### D.4.1  COMPOUND CONFIGURATIONS

As illustrated in Figure 9, we explore how different configurations of QuIC Adapters perform on the STS-B dataset - an illustration of Table 4 in the main text. We note that the presence of $\mathcal{C}_1$ adapter with higher orders show the best performance while giving significant parameter reductions compared to *only* having higher order adapters ($\mathcal{C}_2$ or $\mathcal{C}_3$).

### D.4.2  ORTHOGONALITY

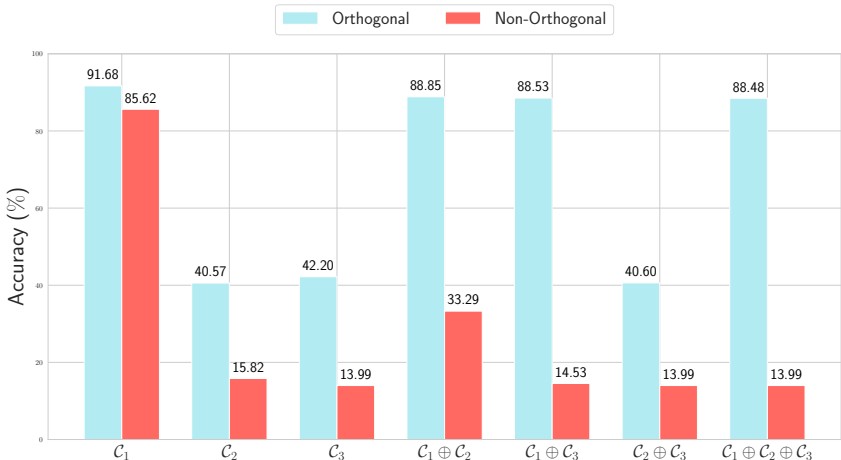

Figure 10: Impact of enforcing orthogonality in adapters, for different compound configurations using STS-B.

To better understand the impact of keeping the adapter parameters orthogonal, we reran the experiments on STS-B but without cayley parameterization. The results are compared with their orthogonal counterpart in Figure  10.

Table 6: STS-B performance comparison for orthogonal vs non-orthogonal implementations, for different compound configurations. The best performing option is in bold.

| Configuration | Orthogonal | Non-Orthogonal |
|---|---|---|
| $\mathcal{C}_1$ | **91.68** | 85.62 |
| $\mathcal{C}_2$ | **40.57** | 15.82 |
| $\mathcal{C}_3$ | **42.20** | 13.99 |
| $\mathcal{C}_1 \oplus \mathcal{C}_2$ | **88.85** | 33.29 |
| $\mathcal{C}_1 \oplus \mathcal{C}_3$ | **88.53** | 14.53 |
| $\mathcal{C}_1 \oplus \mathcal{C}_2 \oplus \mathcal{C}_3$ | **88.48** | 13.99 |

### D.4.3  CONSTRUCTING ADAPTERS FROM ALTERNATE OPERATIONS ON MINORS

We also reran the experiments on STS-B with different operations on the minors as referred to in the main text. The results are compiled in Figure 11.

### D.4.4  RANK AND MULTI-ADAPTER ANALYSIS

We delve into the impact of varying rank options and the number of adapters on the performance of different compound patterns on the STS-B dataset. For each pattern, we evaluate the average accuracy achieved with different rank options (4, 8, 16) and varying numbers of adapters (1 and 4). Additionally, we consider the number of parameters associated with each configuration to assess parameter efficiency alongside performance. We find that in terms of absolute performance, $\mathcal{C}_1 \oplus \mathcal{C}_2$

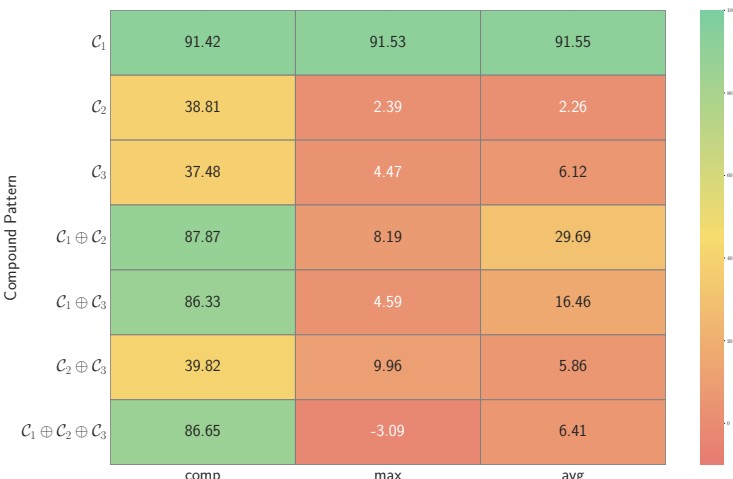

Figure 11: STS-B performance comparison across different operations and compound combinations. `max` and `avg` denotes taking the element wise maximum and average of the minors respectively compared to taking the determinant (`comp`)

with 4 adapters with rank $r = 4$ is the best adapter, however - an optimal tradeoff between high accuracy and low parameter count is achieved with $\mathcal{C}_1 \oplus \mathcal{C}_2$ with only 1 adapter with rank $r = 4$. For this reason, we use the configuration $\mathcal{C}_1 \oplus \mathcal{C}_2$ for the majority of the experiments in the main text.

Table 7: Impact of Rank $r = {}^d/b$ and number of adapters. The best performing configuration in absolute performance is in bold. The results are also visualized in Figure 12.

| Compound Pattern | # Adapters | Rank, $r = {}^d/b$ | Avg Accuracy (%) | Parameters (K) |
|---|---|---|---|---|
| $\mathcal{C}_1 \oplus \mathcal{C}_2$ | 1 | 4 | 90.22 | 35.4 |
| | | 8 | 89.38 | 33.2 |
| | | 16 | 89.25 | 31.9 |
| | 4 | **4** | **91.39** | **139.4** |
| | | 8 | 90.60 | 130.6 |
| | | 16 | 90.23 | 125.2 |
| $\mathcal{C}_1 \oplus \mathcal{C}_3$ | 1 | 4 | 88.51 | 12.4 |
| | | 8 | 89.39 | 16.3 |
| | | 16 | 89.01 | 19.6 |
| | 4 | 4 | 89.61 | 47.2 |
| | | 8 | 90.19 | 62.98 |
| | | 16 | 89.11 | 76.03 |
| $\mathcal{C}_1 \oplus \mathcal{C}_2 \oplus \mathcal{C}_3$ | 1 | 4 | 88.25 | 10.4 |
| | | 8 | 89.13 | 13.1 |
| | | 16 | 88.96 | 14.6 |
| | 4 | 4 | 89.89 | 39.2 |
| | | 8 | 89.02 | 49.9 |
| | | 16 | 89.15 | 56.1 |

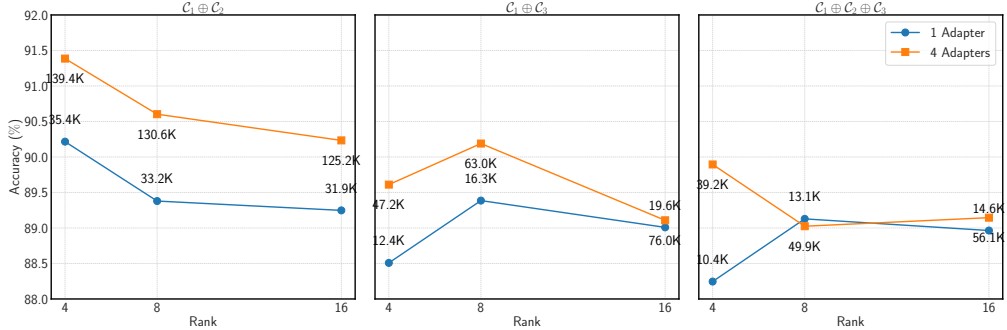

Figure 12: Relationship between rank, $r := {}^d/b$, number of adapters, and accuracy across compound configurations

Figure 12 complements the Table 7 by visually illustrating the trends in accuracy relative to rank and the number of adapters for each compound pattern. The plot highlights the positive correlation

between rank and accuracy, as well as the benefits of employing multiple adapters in enhancing model performance.

# E  HYPERPARAMETERS AND EXPERIMENTAL DETAILS

We report the key hyperparameters and experimental settings used across all benchmarks.All experiments were conducted on a single NVIDIA A80 GPU. Full code available in our anonymised repository: `https://anonymous.4open.science/r/quic-adapters-D41E/README.md`.

## E.1  GLUE BENCHMARK

We evaluate on a subset of GLUE tasks: SST-2, CoLA, MRPC, and STS-B. Table 8 details the main hyperparameters for DeBERTaV3-base. All models are finetuned with AdamW optimizer and linear learning rate decay.

Table 8: Hyperparameters for GLUE (DeBERTaV3-base)

|                     | SST-2 | CoLA | MRPC | STS-B |
|---------------------|-------|------|------|-------|
| Batch Size          | 32    | 32   | 32   | 32    |
| # Epochs            | 2     | 5    | 14   | 11    |
| Learning Rate       | 2e-4  | 4e-4 | 9e-4 | 7e-4  |
| Dropout             | 0.1   | 0.05 | 0.1  | 0.1   |
| Max Sequence Length | 128   | 64   | 320  | 128   |

## E.2  VTAB-1K

We report results on five representative VTAB-1K tasks: CIFAR100, Pets, SVHN, Resisc45, and DMLab. All experiments use Adam optimizer and cosine learning rate schedule. The primary hyperparameter is the initial learning rate, set per task as in Table 9.

Table 9: Learning Rates for VTAB-1K Tasks

| Dataset  | Learning Rate |
|----------|---------------|
| CIFAR100 | 8e-4          |
| Pets     | 3e-4          |
| SVHN     | 3e-3          |
| Resisc45 | 5e-4          |
| DMLab    | 2e-3          |

## E.3  MATH10K

For MATH10K experiments, we use a batch size of 4, AdamW optimizer, and a linear learning rate scheduler with an initial rate of 3e-4.

## E.4  DROP

On DROP, we set batch size to 4, use AdamW optimizer, linear scheduler, and a learning rate of 1e-4.

## E.5  CODE AND REPRODUCIBILITY

All code to reproduce QuIC adapters is available at: `https://anonymous.4open.science/r/quic-adapters-D41E/README.md`.

