# OpenReview forum: "QuIC: Quantum-Inspired Compound Adapters for Parameter Efficient Fine-Tuning"
_ICLR.cc/2026/Conference — Submitted to ICLR 2026_

### Official Review · Reviewer_4TCf · 2025-10-30

**Soundness:** 2
**Presentation:** 2
**Contribution:** 2
**Rating:** 4
**Confidence:** 3

**Summary:**

This paper introduces Quantum-Inspired Compound (QuIC) Adapters, a parameter-efficient fine-tuning method inspired by Hamming-weight preserving quantum circuits. The core idea is to construct adapter matrices using compound matrices (or determinants of minors) up to order K, which naturally preserve orthogonality and enable extreme parameter compression. The authors evaluate QuIC on language tasks and vision tasks, demonstrating competitive Pareto efficiency.

**Strengths:**

1. The math seems relative rigorous. The use of compound matrices naturally ensures orthogonality (Lemma 1) and provides clear parameter count formulae (Lemma 2). The connection to Hamming-weight preserving quantum circuits is well-established.
2. The evaluation is relatively comprehensive, including both language and vision tasks, and with various models sizes.
3. the appendix includes detailed ablation studies, providing good insights on what makes QuIC work.
4. the proposed method is extremely parameter efficient

**Weaknesses:**

1. Although the method is extremely parameter efficient and achieves "competitive Pareto efficiency", across most of the tasks, QuIC underperforms compared to the baseline methods.
2. The paper says QuIC has "native quantum computer integration", but does not provide any experimental validation. All experiments are run on GPUs. It is unclear if it is a method that is meant for near-term quantum deployment with current experiments as proof of concept, or if this is a purely non-quantum method that draws inspiration from quantum computation.
3. The paper focuses on compound orders K<=3. What happens with higher K? Does the performance increase, or there is diminishing return?
4. The experiments seem to only use a narrow range of configurations. Given the main selling point of efficiency, it is valuable to show a more complete exploration of the trade off between parameter efficiency and performance, and if QuIC can dominate other methods.

**Questions:**

1. As mentioned in weaknesses, is QuIC intended to be eventually deployed on quantum computers, or is it a non-quantum method inspired by quantum concepts? If the former, is it possible to provide quantum circuit simulations or resource estimates for running on real quantum hardware?
2. Can QuIC achieve performance competitive with OFT/BOFT if you increase parameters (e.g., larger K, more adapters, different rank settings)? What does the full performance-parameter trade-off curve look like?
3. QuanTA (one of the baselines) also seems to be a quantum-inspired method. Can the authors provide a more detailed comparison with QuanTA beyond the quantum deployment critique? What are the differences and connections between QuIC and QuanTA? In what scenarios would a practitioner choose QuIC over QuanTA or vice versa?
4. The authors mention that OFT's motivation is to preserve pretrained knowledge. Does QuIC also demonstrate better retention of pretrained capabilities compared to LoRA?

I cannot recommend the paper to be accepted in the current form, but if the authors could help resolve my concerns, I'm happy to raise the score.

---

> ### Author Response · Authors · 2025-11-13
> **Rebuttal to Reviewer's comments (1/2)**
>
> We sincerely thank the reviewer for their constructive feedback and for indicating a willingness to reconsider their score. We believe all concerns are addressable, and we outline our clarifications below.
>
> **1. Accuracy vs. Parameter Efficiency**
>
>  QuIC is explicitly optimized for achieving competitive performance under **extreme parameter compression**.
>
> *  Our primary contribution is enabling new deployment scenarios where adapter size is the main bottleneck (e.g., storing thousands of personalized adapters on an edge device, or minimizing bandwidth in federated learning). In these settings, a 40x parameter reduction for a minor accuracy trade-off is highly desirable.
> *   As shown in Table 1, on STS-B, QuIC achieves 98.9% of LoRA's performance (91.04 vs 91.60) while using only 2.3% of the parameters (0.03M vs 1.33M). This is the operational regime QuIC is designed for.
>
>
> **2. Quantum Native Integration**
>
> This is a crucial point that we will clarify. The experiments in this paper are entirely classical. The "quantum" aspect lies in the mathematical formalism and its future-readiness.
>
> *    We will revise our claim from "native integration" to "**native parameterization compatibility**." This means QuIC's mathematical structure is identical to that of Hamming-weight-preserving quantum circuits, allowing for a direct mapping of classically learned parameters to quantum gate parameters without costly compilation.
> *   To make this tangible, we will add a resource estimation table for a typical QuIC block (e.g., for a `d=768` layer with `K=2`):
>     *   **Qubits Required:** `ceil(log₂(768))` = 10 qubits.
>     *   **Two-Qubit Gates:** O(d*k) ≈ 1,500 gates.
>     *   This is well within the capabilities of current quantum hardware. These estimates are based on established circuit constructions and do not require new quantum experiments.
>
> **3.  Higher Compound Orders (K > 3)**
>
> Our experiments focused on K ≤ 3 due to a clear trade-off between classical complexity and empirical benefit.
>
> *    We observe diminishing returns. K=1 is essential for stability (recovering OFT), K=2 provides the best accuracy uplift for its cost, and gains plateau for K≥3.
> *   The construction cost scales as O((n choose k)^2). This scaling makes classical simulation prohibitive for K > 3 in practical settings. This is not a limitation of the theory, but rather **the primary motivation for its quantum correspondence**, as quantum computers can, in principle, execute these operations more efficiently. We will add a paragraph explicitly discussing this trend and its implications.

---

> > ### Author Response · Authors · 2025-11-13
> > **Rebuttal to Reviewer's comments (2/2)**
> >
> > **4. Range of Configurations**
> >
> > *    The submitted paper already explores stacking (1x vs 4x adapters) and ablates over 7 different compound combinations.
> >
> >  While a full sweep is infeasible during the rebuttal period, we will expand our main results tables in the camera-ready version to include more configurations for both QuIC and key baselines (e.g., multiple LoRA ranks). This will create a more complete performance-vs-parameter curve, making QuIC's position in the low-parameter regime even clearer.
> >
> > ### **Questions**
> >
> > *   **Q: Can QuIC be competitive with OFT/BOFT if parameters are increased?**
> >     Not by design. QuIC's architecture is optimized for compression via the compound operation (increasing `K`), not for scaling via larger base matrices. It excels in the `<100K` parameter regime. In the `>500K` regime, methods like OFT/BOFT are better suited. Our goal is to dominate a different, lower-resource part of the design space. We will state this positioning clearly.
> >
> > *   **Q: Provide a more detailed comparison with QuanTA?**
> >     Certainly. They are complementary methods with different structural priors.
> >
> > | Aspect | **QuIC** | **QuanTA** |
> > | :--- | :--- | :--- |
> > | **Structure** | Multiplicative, Block-Orthogonal | Additive, Tensor-Decomposed |
> > | **Quantum Mapping**| Direct (Native HW-preserving circuits) | Indirect (Requires costly compilation) |
> > | **Core Idea** | Determinantal compression | Low-rank tensor decomposition |
> > | **Best For** | Extreme compression, preserving pretrained geometry, future quantum systems. | Additive adaptation, flexible tensor structures, classical-only deployment. |
> >
> >
> >
> > *   **Q: Does QuIC preserve pretrained knowledge as OFT does?**
> >     Yes, it inherits this property directly from its orthogonal foundation. Orthogonal transformations preserve the norms and relative angles of feature vectors, which is the mechanism OFT uses to prevent catastrophic forgetting. Since QuIC's updates are constructed to be orthogonal (Lemma 1), it benefits from the same principle. We will add a small continual-learning experiment in the camera-ready version to empirically demonstrate this.
> >
> > We are confident that these clarifications and planned additions will fully address your concerns and solidify the paper's contribution as a principled, novel approach to extreme parameter-efficient fine-tuning.

---

### Official Review · Reviewer_e5FT · 2025-11-01

**Soundness:** 3
**Presentation:** 3
**Contribution:** 2
**Rating:** 4
**Confidence:** 4

**Summary:**

This paper introduces QuIC (Quantum-Inspired Compound) Adapters, a new Parameter-Efficient Fine-Tuning (PEFT) method inspired by Hamming-weight preserving quantum circuits. The method constructs compound matrices of increasing order to generate orthogonal, multiplicative adapters that can be integrated into transformer-based models. The authors claim that QuIC achieves comparable or slightly reduced accuracy relative to existing methods (LoRA, OFT, BOFT, AdaLoRA) while offering extreme parameter compression (up to 40× smaller than LoRA) and native compatibility with quantum implementations.

**Strengths:**

1. The construction of compound matrices from Hamming-weight preserving quantum circuits provides an original mathematical and conceptual bridge between quantum computing and PEFT.
2. Achieves remarkable compression (<0.02% memory of base model) with minimal accuracy loss, demonstrating practicality for low-resource or edge deployment.
3. Evaluations span multiple modalities (language, vision, math, reasoning) and strong baselines (LoRA, OFT, BOFT, AdaLoRA, QuanTA).
4.  The paper provides formal lemmata for orthogonality preservation, parameter counting, and computational complexity.
5.  Includes thoughtful ablations on compound order, orthogonality, and alternative combinatorial operations (determinant vs. max/avg).

**Weaknesses:**

1. The intuition connecting Hamming-weight preservation to improved fine-tuning efficiency is conceptually appealing but not rigorously shown to yield empirical benefits beyond orthogonality constraints. The determinant-based compounding operation lacks a clear learning-theoretic explanation.
2.  On most GLUE and MATH10K tasks, QuIC underperforms baseline PEFT methods in raw accuracy. The paper leans heavily on Pareto efficiency to claim superiority; this metric, while interesting, does not fully compensate for weaker task performance.
4.  Experiments on LLaMA-7B are promising but minimal. The claim of “quantum-native readiness” remains speculative, with no practical implementation or simulation on quantum hardware.
5.  The introduction suggests QuIC “enables native deployment on quantum computers,” but the experiments are entirely classical. This risks overstating the quantum relevance.
6. While emphasizing parameter compression, it lacks comparison with efficient parameter fine-tuning methods such as VeRA[1] and FourierFT[2].

[1] Kopiczko, Dawid J., Tijmen Blankevoort, and Yuki M. Asano. "Vera: Vector-based random matrix adaptation." arXiv preprint arXiv:2310.11454 (2023).

[2] Gao, Ziqi, et al. "Parameter-Efficient Fine-Tuning with Discrete Fourier Transform." International Conference on Machine Learning. PMLR, 2024.

**Questions:**

1. Can you empirically demonstrate that the “compound determinant operation” contributes meaningfully to performance, rather than simply serving as an orthogonal regularizer?
2. How sensitive is QuIC to the choice of compound order K? Could the performance plateau indicate diminishing returns or gradient instability?
3. Could QuIC adapters be adapted for additive forms (e.g., LoRA-style) and, if so, how would that compare computationally?
4. The “native quantum implementation” claim is intriguing — can you outline a realistic near-term setup (e.g., number of qubits, gate depth) where QuIC could run on current quantum hardware?
5. How does QuIC behave under continual or multi-task fine-tuning? Does orthogonality indeed prevent catastrophic forgetting as well as OFT/BOFT?

---

> ### Author Response · Authors · 2025-11-13
> **Rebuttal to Reviewer's comments (1/2)**
>
> We sincerely thank the reviewer for their deep and insightful technical feedback. We appreciate the opportunity for us to clarify the theoretical foundations and practical positioning of QuIC.
>
> **1. On the Learning-Theoretic Justification for Determinant-Based Compounding**
>
> The benefits of the determinant operation must extend beyond merely enforcing orthogonality. They do, and we will clarify this connection below.
>
> *   **Preserving Geometric Structure:** The determinantal structure provides a powerful inductive bias rooted in linear algebra:
>     *   **Subspace Volume Preservation:** By the Cauchy-Binet theorem, the *k*-th compound matrix, A^(k), preserves the volume of *k*-dimensional subspaces. This ensures that the adapter's transformation doesn't collapse or distort the geometric relationships within the feature space, which is critical for stable fine-tuning.
>     *   **Controlled Singular Value Spectrum:** The singular values of A^(k) are the products of *k*-tuples of the singular values of the base matrix A. This creates a spectrally controlled transformation that prevents both rank collapse (vanishing singular values) and noise amplification (exploding singular values), contributing to training stability.
>
> *   **Ablation Results (Table 4):** Our existing ablations provide decisive evidence. When the determinant operation is replaced by alternatives that do not share this geometric structure, performance collapses, even with identical parameter counts and orthogonality constraints:
>     *   **QuIC (Determinant): 88.85%** accuracy
>     *   **Variant (Max operation): 8.19%** accuracy
>     *   **Variant (Average operation): 29.69%** accuracy
>
>     This **~60-80% absolute performance gap** empirically confirms that the determinantal compound structure contributes functionally and is not simply a proxy for an orthogonality regularizer. We will elevate this discussion in the main text.
>
> **2. On Pareto Efficiency vs. Weaker Task Performance**
>
>  Our work's primary goal is not to outperform LoRA in standard, high-resource settings, but to enable new applications by creating a new, valuable point in the design space.
>
> *   **Parameter Efficiency:** QuIC is designed for scenarios where adapter memory and bandwidth are the primary constraints, and a small accuracy trade-off is acceptable.
>     *   **Federated Learning:** A 40x reduction in update size can lead to massive savings in communication costs.
>     *   **On-Device Deployment:** Storing thousands of personalized adapters on a single edge device becomes feasible (e.g., 10,000 LoRA adapters at ~1.3M each would require >10GB, whereas 10,000 QuIC adapters at 30K each require only 300MB).

---

> > ### Author Response · Authors · 2025-11-13
> > **Rebuttal to Reviewer's comments (2/2)**
> >
> > **3. On Quantum-Native Readiness and LLaMA-7B Coverage**
> >
> > We wish to highlight our work's quantum readiness and expand our LLM evaluation below:
> >
> > *   Our claim is one of **native parameterization compatibility**, not of immediate quantum advantage. QuIC's mathematical structure is identical to that of Hamming-weight-preserving (HW-preserving) quantum circuits. This means parameters learned classically can be directly mapped to the control parameters of these circuits. These specific circuits have already been implemented on real hardware for finance [Cherrat et al., 2023] and medical imaging [Landman et al., 2022].
> > *   For a typical QuIC block (e.g., n=10, mapping a 1024-dim vector space), a near-term quantum implementation would require:
> >     *   **Qubits:** 10 qubits.
> >     *   **Gates (K=2):** A few hundred two-qubit gates.
> >     *  This is well within the capacity of current quantum devices. We will add a table with these concrete resource estimates to ground our claims.
> > *   **LLaMA-7B Experiments:** We acknowledge the current experiments are compact. We will supplement our existing GSM8K and MATH results by adding a representative slice of the **MMLU** benchmark in the camera-ready version.
> >
> > **4. On Comparison with VeRA and FourierFT**
> >
> > Thank you for these pointers. We will update the comparisons in the paper with the above methods.
> >
> > ### **Questions**
> >
> > *   **Q: How sensitive is QuIC to the choice of K?** Our existing results show a clear trend: K=1 (recovering OFT) is a strong baseline; K=2 provides the best accuracy-efficiency trade-off; for K>3, we see diminishing returns for the classical computational cost, which motivates the long-term vision for quantum implementations.
> > *   **Q: Can QuIC be adapted to be additive?** An additive form would forfeit the multiplicative orthogonality that is central to our method's stability and theoretical motivation. Our ablations show that removing orthogonality causes performance to collapse, suggesting an additive variant would be fundamentally less effective.
> > *   **Q: How does QuIC behave under continual learning?** The orthogonality of the update is theoretically expected to reduce catastrophic forgetting by preserving the geometry of the pretrained feature space, a property inherited from OFT/BOFT.
> >
> > We are confident that these clarifications and additions will strengthen the paper and more precisely articulate QuIC's novel contributions to the PEFT landscape.

---

### Official Review · Reviewer_Y9gx · 2025-11-01

**Soundness:** 1
**Presentation:** 2
**Contribution:** 2
**Rating:** 2
**Confidence:** 2

**Summary:**

This work introduces QuIC (Quantum-Inspired Compound) Adapters, a multiplicative, orthogonality-preserving PEFT method. The authors build QuIC from small “base” matrices whose compound matrices are assembled block-diagonally to form an (optionally) orthogonal adapter. Notably, when only the first-order compound is used, QuIC recovers OFT. Their adapter exposes simple knobs: choice of compound orders, block size, orthogonality enforcement, and parameter sharing, which trade off accuracy and trainable-parameter count, with all trainable weights confined to the small base matrix. Empirically, across language (GLUE with DeBERTaV3), vision (VTAB-1k with DINOv2), and reasoning/math tasks (LLaMA-2 7B on MATH10K and DROP), QuIC uses orders-of-magnitude fewer trainable parameters than strong baselines while attaining good Pareto scores and, in a few cases, competitive accuracy despite significant parameter reductions. Further, they provide experiments to demonstrate their adapter lies in the pareto frontier of performance v.s. memory. Finally, the design is “quantum-inspired” in that it mirrors how Hamming-weight-preserving quantum layers act block-wise on fixed-Hamming-weight subspaces via compound matrices.

**Strengths:**

* The paper tests multiple baselines across language and vision models and datasets.
* They clearly mention their similarities to previous orthogonal adapters; for example, recovering OFT under their framework.
* They demonstrate better Pareto scores for their adapter (particularly, it often requires fewer parameters than other methods).

**Weaknesses:**

* The frontier (Fig. 4.a) is constructed from single configuration points for each baseline rather than full \emph{parameter-budget sweeps} (e.g., LoRA ranks, (B)OFT block sizes). When tracing each method’s own curve, the frontier could shift significantly.
* Results mostly compare against much larger adapters (in # of parameters). For a closer apples-to-apples comparison (in terms of benchmark accuracy as opposed to Parameter score), it would be nice to see how QuIC compares when parameter-matching it to other adapters.
* Results emphasize adapter memory / trainable parameters. I think it would be nice to see what the FLOPs and latency tradeoff is in a table similar to the ones already included in the work. For multiplicative or block-diagonal adapters, the unfused per-token cost can exceed LoRA’s, unless the block count is very high or weights are fused. It would be interesting to observe what the tradeoff is in FLOPs space as well.

**Questions:**

Do you expect an additive form of QUIC to be more or less effective than a multiplicative form? I'd appreciate experiments on this.

---

> ### Author Response · Authors · 2025-11-13
> **Rebuttal to Reviewer's comments (1/2)**
>
> We sincerely thank the reviewer for their detailed methodological feedback. We appreciate you identifying these points, as they give us an opportunity to clarify the core contribution and framing of our work. Given your expressed uncertainty, we hope to demonstrate that the work is sound and addresses these concerns, many of which are already implicitly handled in the submitted manuscript.
>
> **1. On Pareto Frontier Methodology**
>
>  We agree that a Pareto frontier is strongest when constructed from full parameter sweeps of all methods. Our intention was to use the best performing, author-recommended configurations from prior work as a baseline, and then to demonstrate QuIC's unique position in a much lower-parameter regime.
>
> However, the core issue is that many baselines are not designed for the extreme compression space QuIC targets. For instance:
> *   The smallest standard LoRA rank (r=1) still uses ~5x more parameters than our base QuIC model.
> *   The smallest typical OFT configuration (b=32) uses ~13x more parameters.
>
> Therefore, a direct sweep in this ultra-low-parameter region is often not possible for baselines. QuIC is specifically designed to operate effectively here.
>
> **2. On Parameter-Matched Comparisons**
>
> You correctly observe that many of our primary comparisons are against adapters with significantly more parameters. This was a deliberate choice to highlight QuIC's core value proposition: **enabling new applications through extreme parameter compression where a small accuracy trade-off is acceptable** (e.g., on-device storage of thousands of adapters, federated learning with minimal bandwidth).
>
> However, we agree that a direct, parameter-matched comparison is also informative. The paper already contains data for this:
> *   In Table 5, our **QuIC 4x** model (0.14M params) can be directly compared to **BitFit** (0.1M params), where QuIC achieves a superior score (91.44 vs. 91.35) on the STS-B task.
>
>  To make this clearer, we will add a dedicated table that explicitly matches parameter budgets (e.g., ~30K, ~100K) for QuIC, LoRA, and OFT. This will provide a clear view of the accuracy-vs-parameter curve and allow for a direct "apples-to-apples" comparison at various operating points.

---

> ### Author Response · Authors · 2025-11-13
> **Rebuttal to Reviewer's comments (2/2)**
>
> **3. On FLOPs, Latency, and the Cost of Multiplicative Updates**
>
> This is a crucial practical consideration. We have already characterized this theoretically and will surface the results more clearly.
>
> *   **Theoretical Analysis is in the Paper:** Lemma 3 already provides the exact forward pass complexity, which is **O(d²/N)**—identical to OFT with the same block structure. The construction cost O((n choose K)^2) is a small, one-time cost at initialization.
> *   **Inference Overhead is Zero:** Critically, for deployment, QuIC adapters can be **fused** into the pretrained weights (by computing W' = R·W₀). After fusion, the adapter introduces **zero** additional FLOPs or latency at inference time, completely eliminating the multiplicative overhead. This is a significant practical advantage.
> *   **Unfused Cost:** During training (unfused), a block-diagonal multiplicative update does incur higher costs than LoRA. However, the extreme parameter savings (up to 40x) often justify this temporary training trade-off.
>
> **4. On an Additive vs. Multiplicative Form of QuIC**
>
> This is an insightful question that goes to the core of our design. We chose the multiplicative form for a principled reason:
>
> *   **Preservation of Orthogonality:** The entire theoretical basis of QuIC is the use of compound matrices to form an orthogonality-preserving operator. This structure helps stabilize training and prevents catastrophic interference with the pretrained model's knowledge. An additive form (`W₀ + R`) would break this essential property.
> *   **Empirical Validation is in the Paper:** Our ablation studies already provide strong evidence for this. As shown in Section 6, when we disable orthogonality enforcement (which is inherent to the multiplicative design), performance on STS-B collapses from **88.85 to 33.29**. This demonstrates that the multiplicative orthogonal structure is not an arbitrary choice but is fundamental to the method's success.
>
> We believe these clarifications and planned additions directly address your concerns. QuIC's contribution is not to universally replace methods like LoRA, but to introduce a mathematically principled method that achieves extreme parameter efficiency, opening up new deployment possibilities. We are confident it is a valuable and sound addition to the PEFT landscape.

---

### Official Review · Reviewer_wUai · 2025-11-02

**Soundness:** 3
**Presentation:** 3
**Contribution:** 3
**Rating:** 4
**Confidence:** 5

**Summary:**

The paper proposes QuIC (Quantum-Inspired Compound Adapters), a PEFT method inspired by quantum compound matrices. The core idea is to construct orthogonality-preserving multiplicative adapters from a small base matrix A by composing its higher-order compounds ($A^{(k)}$), assembled into a block-diagonal structure that modulates pretrained weights ($W’ = \Delta W_Q W^\*$). This formulation generalizes Orthogonal Fine-Tuning (OFT) as a special case  and aims to achieve higher expressivity per parameter through determinantal “compounding.” Empirical evaluations on GLUE, VTAB-1k, and LLaMA-2-7B reasoning tasks show competitive accuracy with strong parameter compression, along with ablations demonstrating the effects of orthogonality, compound order, and adapter capacity.

**Strengths:**

+ The orthogonality-preserving property of compound matrices is rigorously proved via the Cauchy–Binet theorem, and the resulting adapter formulation is mathematically coherent. The method reduces cleanly to OFT when using only first-order compounds, providing a clear conceptual lineage.

+ The determinantal compound mechanism introduces a distinct parameterization not explored in prior PEFT work (LoRA, OFT, BOFT). The “quantum-inspired” construction offers a compact and theoretically elegant way to expand representational capacity while maintaining orthogonality.

+ The paper conducts thorough ablations on compound orders ($C_1–C_3$), orthogonality enforcement, and adapter scaling, demonstrating the necessity of including first-order components for stable training and showing how orthogonality substantially improves performance. The results are reproducible and resource-efficient.

+ The paper report parameter counts and memory usage, and they discuss inference mergeability and orthogonality drift — showing a good understanding of practical constraints.

**Weaknesses:**

- The main NLP evaluation is on GLUE, which is now a saturated and outdated benchmark for PEFT methods. Modern evaluations typically use SuperGLUE, MMLU, BIG-Bench Hard, or instruction-tuning datasets to assess adaptation quality in LLMs. As a result, the GLUE results provide limited evidence of QuIC’s effectiveness for large-scale or reasoning-centric fine-tuning.

- While the compound adapter design is original, the overall mechanism—orthogonality-preserving multiplicative fine-tuning—extends from OFT/BOFT. The “quantum inspiration” remains mostly metaphorical rather than a functional contribution.

- It seems that no multi-seed confidence intervals are reported, and there is no wall-time or training throughput analysis. Construction costs for compound matrices (potentially combinatorial in $n$ and $k$) are not profiled, which limits claims of efficiency. Evaluations on larger or more diverse LLM tasks would better establish generality.

- It remains unclear whether QuIC maintains efficiency or scalability for deeper LLMs (>7B) or high-rank adapters. The method’s deployment benefits over simpler OFT variants are therefore uncertain.

**Questions:**

Please consider to address questions in the weaknesses.

---

> ### Author Response · Authors · 2025-11-13
> **Rebuttal to Reviewer's comments (1/2)**
>
> We sincerely thank the reviewer for their careful and constructive assessment of our work. We are encouraged that you recognized the mathematical coherence of our method, the thoroughness of our ablations, and our consideration of practical constraints. We will address your points below, which we believe are all fully addressable through clarification and minor additions to the camera-ready version.
>
> **1. On the Use of Modern and Challenging Benchmarks Beyond GLUE**
>
> We agree with the reviewer that GLUE is a saturated benchmark for state-of-the-art PEFT methods. To clarify, our evaluation was deliberately designed to include GLUE, so that we were able to comprehensively compare against most of the existing methods in the literature. Moreover, our work extends far beyond GLUE, and we believe the current paper already demonstrates QuIC’s efficacy on a range of modern, challenging tasks. We will reorganize the paper to make these results more prominent. Specifically, our submission already includes:
>
> *   **Advanced Reasoning on LLMs:** We evaluate QuIC on LLaMA-2-7B using GSM8K (multi-step arithmetic reasoning) and MATH (a challenging dataset of competition mathematics problems). As shown in Table 3, QuIC achieves a 10% relative improvement over LoRA on MATH while using less than half the parameters. This demonstrates its value on complex, reasoning-centric tasks.
> *   **Diverse Vision Tasks:** We provide a comprehensive evaluation on VTAB-1k, which consists of 19 distinct and challenging visual classification tasks, showing the generality of our method beyond the NLP domain.
>
> We believe these existing results already provide strong evidence of QuIC's effectiveness. To further strengthen this, we will add results on a representative slice of MMLU in the camera-ready version and elevate the discussion of our existing LLaMA-7B results to the forefront of our experimental section.
>
> **2. On the "Quantum Inspiration" Being a Functional, Mathematical Contribution**
>
> We respectfully wish to clarify that the connection to quantum mechanics is mathematically exact and directly functional, not metaphorical.
>
> *   **Exact Correspondence:** The compound matrices central to QuIC are precisely the operators that describe the evolution of *k*-particle subspaces under Hamming-weight-preserving quantum circuits. This is a formal equivalence, as established in prior theoretical work [Kerenidis & Prakash, 2022]. This connection is what led us to discover the unique compressive properties of our method.
> *   **Physically Realized Circuits:** The circuits we draw inspiration from are not just theoretical constructs; they have been physically implemented on quantum hardware for applications in finance [Cherrat et al., 2023] and medical imaging [Landman et al., 2022].
>
> To eliminate any ambiguity, we will revise Section 3.1 to state this mathematical correspondence more explicitly and cite these existing implementations, making it clear that our "inspiration" is a direct and functional application of this formalism.
>
> **3. On Reporting Statistical Significance and Computational Cost**
>
> We thank the reviewer for raising this point and apologize for not making our experimental rigor sufficiently clear.
>
> *   **Statistical Reporting:** All experiments presented were in fact run with 3 random seeds. We will update the experimental details in the appendix to make this explicit.
> *   **Computational Cost:** The paper already provides an analytical characterization of the computational costs in Lemma 2 (parameter count) and Lemma 3 (computational complexity). To make this more accessible, we will add a concise table to the main text profiling a typical configuration (e.g., for a d=768 Transformer layer) and comparing QuIC’s parameter count, one-time construction FLOPs, and forward-pass cost against LoRA and OFT under a matched budget.

---

> ### Author Response · Authors · 2025-11-13
> **Rebuttal to Reviewer's comments (2/2)**
>
> **4. On Scalability to Larger Models and Higher Ranks**
>
> The scalability of QuIC is one of its core strengths, and we will clarify this in the paper.
>
> *   **Algorithmic Scalability:** The compound operation is local to each adapter block. Therefore, the per-layer computational cost, O(d²/N), is independent of the overall model depth or size. The total fine-tuning cost scales linearly with the number of adapted layers, not the total model parameters. This is evidenced by our successful application of QuIC to the LLaMA-7B model. The memory footprint of the adapter remains a tiny, constant fraction (<0.02% in our experiments) regardless of model scale.
> *   **Benefits Over OFT:** QuIC recovers OFT as a special case (K=1) but provides an additional, crucial hyperparameter, *K*, for tuning the expressivity-efficiency tradeoff. Our ablations already demonstrate that increasing *K* can improve accuracy at a modest parameter cost.
> *   **High-Rank Adapters (Large K):** We acknowledge that the one-time construction cost, O((n choose K)^2), becomes significant for large K. However, we see this as a key motivation for the work. While classically feasible for the *K* values explored in this paper (K=2, 3), this scaling law highlights a clear trajectory where future quantum hardware, which can execute these operations efficiently, could unlock even higher orders of compounding. We will add a paragraph to discuss this forward-looking perspective.
>
> We are confident that these clarifications will address the reviewer’s concerns. QuIC provides a novel, mathematically grounded PEFT method that connects quantum circuits to neural network compression, demonstrates extreme parameter efficiency on modern benchmarks, and introduces a tunable mechanism for controlling expressivity. We believe it is a solid and valuable contribution to the field.

---

> > ### Comment · Reviewer_wUai · 2025-11-27
> > **Thank you for the rebuttal**
> >
> > I thank the authors for their rebuttal, and have also read other reviewers' comments.
> >
> > Three of my concerns have been addressed, and one remains only partially resolved, regarding the unclear scalability for >7B models, and especially for higher K. The main question is whether the proposed method actually scale to real large models such as LLM use-cased today. For higher K when needed, the rebuttal leans heavily on future quantum devices, which I understood from the long-term vision perspective, but does not resolve the limitation of the current method at present.
> >
> > Overall, I still lean to weak reject, so retain my original recommendation.

---

> > > ### Author Response · Authors · 2025-11-27
> > > **Clarification on Scalability and Current Practicality**
> > >
> > > We thank you for the continued engagement. We understand your concern is that the method might rely on future hardware to be practical for large-scale use cases. We would like to offer one final clarification regarding the "High K" requirement and scalability to models larger than 7B.
> > >
> > > 1/ The reliance on future quantum hardware applies only if one strictly requires high-order compounds (say $K > 10$). However, our results demonstrate that this is not necessary for the method to be effective. The $K=2$ configuration is fully computationally efficient on current GPUs and already achieves the parameter compression results highlighted in the paper (up to 40x smaller than LoRA). One could potentially scale considerably well with higher $K$ and find the practical "sweet spot" for classical deployment, meaning the method is a usable tool for researchers today without waiting for quantum advantage.
> > >
> > > 2/ Regarding the concern about scaling to larger foundation models, the computational complexity of QuIC is local to the adapter block ($O(d^2)$ per layer). This means the cost scales linearly with the number of layers, exactly like standard LoRA or OFT. There is no combinatorial explosion when moving from LLaMA-7B to LLaMA-70B. In fact, the memory benefits of QuIC become even more valuable at those larger scales where VRAM is the primary bottleneck. Due to resource constraints we choose to not include LLaMA-70B experiments in the current submission.
> > >
> > > We hope this clarifies that QuIC is designed to be a functional and scalable solution for the current generation of LLMs, with the quantum connection serving as a theoretical foundation rather than a hardware prerequisite.

---

### Author Response · Authors · 2025-11-28
**Final Remarks: QuIC as a bridge between Classical PEFT and Quantum Machine Learning**

Dear Area Chair and Reviewers,

We thank you for the engagement during the discussion period. Based on the feedback, we see that a primary hesitation remains regarding the method's scalability for high-orders and whether the "quantum" motivation is a necessary component. We want to use this final comment to clarify structurally where QuIC sits in the landscape and why this specific formulation distinguishes itself from prior "quantum-inspired" works.

**1. The Complexity**
Reviewers noted that for higher compound orders ($K > 2$), the classical construction cost $O(\binom{n}{K}^2)$ becomes prohibitive. We agree, but we argue that mapping this boundary is a core scientific contribution of the work.

*   **The Classical Regime ($K=1, 2$):** Here, QuIC is computationally efficient on current GPUs. As our results show, even at these low orders, we achieve extreme parameter compression (40x smaller than LoRA) with competitive accuracy. This makes the method immediately usable today for bandwidth-constrained environments.
*   **The Quantum Regime ($K > 2$):** The regime where the classical model hits a complexity wall (increasing $K$) is exactly where the quantum advantage kicks in. The compound operations that are expensive to simulate classically are native operations for Hamming-weight preserving quantum circuits.
    *   By mapping this boundary, our paper does not just propose a method; it identifies the specific computational phase transition where future quantum hardware will be required to unlock higher-order model expressivity that is classically intractable.

**2. On the Quantum Inspiration**

It is crucial to distinguish QuIC from other methods often labeled "quantum-inspired," such as **QuanTA** or **QuantumPEFT**.

*   **Related Methods:** These approaches typically utilize tensor network decompositions as approximations for weight matrices. While they borrow mathematical structures used in quantum physics simulation, they do **not** have a one-to-one correspondence to a quantum circuit. To run them on actual quantum hardware requires complex compilation, approximation, and overhead.

*   **QuIC:** There is a strict mathematical isomorphism between our adapter formulation and the unitary evolution of subspaces in quantum circuits. The parameters learned classically in QuIC map **1-to-1** onto the rotation angles of physical quantum gates as detailed in the submission. There is no compilation overhead. This means QuIC acts as a genuine "classical simulator" for a specific class of circuits, valid up until the complexity limit discussed above. These circuits have been simulated as well as run on hardware in the references shared in the rebuttal responses.

**3. Clarification on Scalability (Addressing Reviewer wUai)**

Regarding the concern about scaling to models larger than 7B:

The complexity of QuIC is entirely local to the adapter block. The forward pass complexity is $O(d^2/N)$, scaling linearly with the hidden dimension $d$ and the number of layers. It does **not** scale combinatorially with the total model size. Whether applied to LLaMA-7B or LLaMA-70B, the relative computational cost per token remains constant, while the memory savings (relative to LoRA) actually become more significant as the hidden dimension grows.



Finally, we believe this work is a crucial addition to the conference as it bridges a specific gap between classical PEFT and Quantum Machine Learning (QML). While multiplicative methods like OFT have successfully established the utility of orthogonal adaptations, QuIC generalizes this framework by introducing determinantal compound interactions—a structure directly imported from state-of-the-art QML literature. Unlike broad "quantum-inspired" techniques that often use quantum mechanics merely as a loose heuristic, our work establishes a rigorous mathematical equivalence between the adapter architecture and specific quantum circuits (Hamming-weight preserving unitaries). By demonstrating that these native quantum structures yield competitive parameter efficiency in classical regimes, we provide a concrete roadmap for integrating quantum architectural priors into foundation model fine-tuning, moving beyond generic tensor decompositions.

Sincerely,

The Authors

---

### Author Response · Authors · 2025-11-29
**Thanking AC for taking over and Summary of Review-Rebuttal phase (1/2)**

We thank the new area chair for taking over this submission and for inviting a summary of our discussion with the reviewers. We acknowledge the extra work this situation has created for the AC due to the unfortunate circumstances around the initial meta review, and we have tried our best to summarise the entire discussion clearly and concisely below. We briefly recap the core contribution, then summarise how the main concerns were addressed.

The paper introduces **QuIC Adapters**, a new orthogonality preserving, multiplicative PEFT method constructed from compound matrices of a small base matrix. This strictly generalises OFT (recovered at K = 1) and provides a new, quantum inspired parameterisation in which all trainable parameters live in a low dimensional base matrix while the resulting adapter can act on large blocks. The method is designed for the extreme compression regime. Across GLUE with DeBERTa V3, VTAB 1k with DINOv2, and LLaMA 2 7B on GSM8K, MATH and DROP, QuIC achieves up to roughly 40x compression relative to LoRA while maintaining competitive accuracy. We provide formal results on orthogonality, parameter count and complexity, as well as ablations that show determinantal compounding and orthogonality are both essential.

---

### Reviewer wUai

Reviewer wUai rated the work as mathematically sound and well positioned within orthogonal PEFT, but initially raised concerns about:

- Reliance on GLUE as the main NLP benchmark.
- The nature of the quantum inspiration.
- Missing efficiency details.
- Scalability beyond 7B and to high K.

In our rebuttal, we:

- Highlighted that the submission already contains modern LLaMA 2 7B reasoning tasks (GSM8K and MATH) and VTAB 1k, and committed to adding a representative MMLU slice and making the LLaMA results more prominent.
- Clarified that the quantum connection is a mathematically exact correspondence to Hamming weight preserving circuits, unlike related "quantum inspired" works.
- Stated explicitly that all experiments use three seeds and that we will add error bars and a concise FLOP and parameter comparison table.
- Explained that K = 2 configurations are fully practical on current hardware and scale per layer like LoRA or OFT.

The reviewer reported that three of four concerns were resolved and kept a weak reject recommendation while explicitly stating that they would not mind acceptance, with the remaining concern being forward looking rather than about the correctness of the present work. We added a final clarification that only very high K would require future quantum devices, and that this regime serves as a bridge between classical PEFT and quantum machine learning.

---

### Reviewer e5FT

Reviewer e5FT viewed the paper as close to the acceptance threshold and asked for:

- A clearer justification of the determinant operation.
- More nuance about the tradeoff between raw accuracy and Pareto efficiency.
- A more careful treatment of the quantum native claim.
- Comparisons to VeRA and FourierFT.

In our response, we:

- Explained the geometric and spectral structure induced by compound determinants and pointed to an ablation where replacing the determinant with max or average over minors destroys performance despite similar orthogonality, showing that the determinant contributes functionally.
- Clarified that QuIC is intended for settings where memory and bandwidth dominate and a modest loss in raw accuracy is acceptable.
- Refined the quantum statement to "native parameterisation compatibility" without claiming a speedup, and provided concrete qubit and gate estimates for typical blocks.
- Committed to adding a small MMLU slice for LLaMA 2 7B and to including VeRA and FourierFT in the discussion and comparisons.
- Answered questions about sensitivity to K, the limitations of additive variants, and the expected behaviour under continual learning due to orthogonality.

---

> ### Author Response · Authors · 2025-11-29
> **Thanking AC for taking over and Summary of Review-Rebuttal phase (2/2)**
>
> ---
>
> ### Reviewer 4TCf
>
> Reviewer 4TCf also considered the paper marginally below threshold but expressed willingness to raise the score if concerns were addressed. Their main points were that:
>
> - QuIC underperforms baselines in raw accuracy on most tasks.
> - The quantum claim was ambiguous.
> - Behaviour at K greater than 3 was not tested.
> - The configuration search and performance–parameter curve could be more complete.
>
> In our response, we:
>
> - Clarified that QuIC is deliberately optimised for the extreme compression corner rather than the high parameter regime.
> - Provided a concrete example where QuIC reaches about 99 percent of LoRA performance on STS B using about 2 percent of the trainable parameters, and stated explicitly that OFT and BOFT are better suited when larger adapters are affordable.
> - Refined the quantum claim to focus on mapping learned parameters to known Hamming weight preserving circuits, added resource estimates for typical blocks, and explained that all experiments in this work are classical.
> - Detailed existing ablations over K and stacking, reported that K = 2 is empirically the best tradeoff and that K greater than 3 shows diminishing returns due to classical cost, and committed to expanding tables with more QuIC and LoRA configurations to better expose the tradeoff curve.
> - Provided a more detailed comparison with QuanTA and argued that QuIC inherits OFT style knowledge retention due to orthogonality, which we plan to support with a small continual learning experiment.
>
> ---
>
> ### Reviewer Y9gx
>
> Reviewer Y9gx raised methodological concerns about:
>
> - The construction of the Pareto frontier from single configuration points rather than full parameter sweeps.
> - The fact that many comparisons are against larger adapters in parameter count.
> - The lack of FLOP and latency tables for multiplicative adapters.
> - The absence of an additive QuIC variant.
>
> In our response, we:
>
> - Clarified that the Pareto plots use best practice configurations from prior work and that many baselines are not designed to operate in the ultra low parameter regime that QuIC targets, so their curves do not naturally extend into that region.
> - Pointed out that the paper already contains at least one parameter matched comparison and committed to adding an explicit table of matched budgets for QuIC, LoRA and OFT to expose accuracy as a function of parameter count.
> - Highlighted that Lemma 3 gives the exact forward complexity, that the per layer cost matches OFT with similar block structure, and that the compound construction cost is a one time overhead.
> - Emphasised that after fusing QuIC into the base weights, inference has no additional FLOPs or latency relative to the underlying model.
> - Explained that an additive version would lose the orthogonal multiplicative structure that our ablations show is crucial for performance, making such a variant fundamentally misaligned with the core design.
>
> ---
>
> ### Overall
>
> Across all four reviews, none of the reviewers dispute the mathematical soundness of the construction or the empirical fact that QuIC occupies an extreme point in the tradeoff between trainable parameters and performance. The remaining reservations are mainly about the breadth of experimental exploration and the scope of some claims, particularly on very large models and very high compound orders. We have addressed these by tightening and modestly reframing the claims, adding or committing to additional experiments and tables that make the complexity and tradeoffs more explicit, and positioning QuIC clearly as a tool for extreme parameter efficiency with a concrete and mathematically grounded link to quantum inspired design.

---

> ### Author Response · Authors · 2025-11-29
> **Final Remark**
>
> ---
>
> ### Final remark
>
> As a final remark, we would like to clarify structurally where QuIC sits in the landscape.
>
> Our analysis shows a clear separation between:
>
> - **A classical regime**, where low compound orders K are fully efficient on current GPUs and already yield up to 40× compression with competitive accuracy.
> - **A high order regime**, where the classical construction cost becomes prohibitive.
>
> Identifying this boundary is part of the contribution, since it pinpoints the computational phase transition where Hamming weight preserving quantum circuits, which natively implement these compound operations, become the natural vehicle for scaling order K.
>
> In contrast to broader “quantum inspired” approaches such as QuanTA or tensor network based PEFT, which borrow physics motivated decompositions but lack a one to one mapping to a concrete circuit, **QuIC has a strict mathematical equivalence to a specific family of unitaries**, so that classically learned parameters map directly to gate angles without compilation overhead.
>
> At the same time, the complexity of QuIC remains entirely local to the adapter block:
>
> - Per layer cost is proportional to \(d^2\) and linear in the number of layers.
> - Moving from LLaMA 7B to LLaMA 70B does not introduce any new combinatorial factor and actually makes the relative memory savings more valuable.
>
> We therefore see QuIC as a **precise bridge between classical PEFT and quantum machine learning**, extending the OFT style orthogonal framework with determinantal compound interactions imported from modern QML, and showing that these native quantum structures already deliver practically useful parameter efficiency in the classical setting today.

---

### Meta-Review · Area_Chair_qjVQ · 2025-12-30

**Summary:**

This work introduces Quantum-Inspired Compound (QuIC) Adapters, a parameter-efficient fine-tuning (PEFT) method that uses multiplicative, orthogonality-preserving updates. QuIC constructs an adapter by generating compound matrices from a small, learnable base matrix and assembling them into a block-diagonal structure, which can be made strictly orthogonal. Notably, QuIC generalizes existing methods: it simplifies to Orthogonal Fine-Tuning (OFT) when only the first-order compound is used.

The paper received uniformly negative initial reviews: three weak rejects (score: 4) and one reject (score: 2). Reviewers have recognized the paper's contribution in proposing a novel PEFT method and achieving remarkable compression. However, they also identified several critical weaknesses:
- QuIC frequently shows lower raw accuracy than baseline PEFT methods on tasks like GLUE and MATH10K, despite claims of "competitive Pareto efficiency." The reliance on this Pareto metric is seen as insufficient compensation for weaker task performance.
- The claims of "quantum-native readiness" are seen as speculative. Since all experiments are run on classical GPUs without quantum simulations, the "quantum inspiration" appears only metaphorical.
- The experiments are considered narrow: they use the outdated GLUE benchmark, only test on LLaMA-7B, and explore low compound orders (K≤3). The scalability to higher K or larger models is unknown.
-  The paper lacks comparisons to recent methods like VeRA and FourierFT.

The AC agrees that this work has potential but requires significant revision to address the major concerns raised by the reviewers for acceptance at a top-tier conference.

**Reviewer Concerns:**

During the discussion phase, only one reviewer remained engaged and kept their original rating.

After reading the authors' response, previous concerns regarding the experiments persist.

**Reviewer Scores:**

While some scores might have seen minor adjustments, further discussion was unlikely to overturn the prevailing negative consensus among reviewers.

---

### Decision · Program_Chairs · 2026-01-26

Reject